# Concerted pulsatile and graded neural dynamics enables efficient chemotaxis in *C. elegans*

Eyal Itskovits [1,2], Rotem Ruach[1,2], Alexander Kazakov[3] & Alon Zaslaver[1]

The ability of animals to effectively locate and navigate toward food sources is central for survival. Here, using *C. elegans* nematodes, we reveal the neural mechanism underlying efficient navigation in chemical gradients. This mechanism relies on the activity of two types of chemosensory neurons: one (AWA) coding gradients via stochastic pulsatile dynamics, and the second (AWC[ON]) coding the gradients deterministically in a graded manner. The pulsatile dynamics of the AWA neuron adapts to the magnitude of the gradient derivative, allowing animals to take trajectories better oriented toward the target. The robust response of AWC[ON] to negative derivatives promotes immediate turns, thus alleviating the costs incurred by erroneous turns dictated by the AWA neuron. This mechanism empowers an efficient navigation strategy that outperforms the classical biased-random walk strategy. This general mechanism thus may be applicable to other sensory modalities for efficient gradient-based navigation.

[1] Department of Genetics, The Silberman Institute of Life Science, Edmond J. Safra Campus, the Hebrew University of Jerusalem, Jerusalem, Israel. [2] School of Computer Science and Engineering, the Hebrew University of Jerusalem, Jerusalem, Israel. [3] Edmond and Lily Safra Center for Brain Sciences, the Hebrew University of Jerusalem, Jerusalem, Israel. These authors contributed equally: Eyal Itskovits, Rotem Ruach. Correspondence and requests for materials should be addressed to A.Z.(email: alonzas@mail.huji.ac.il)

Animals heavily rely on efficient detection of chemical cues to guide them in critical processes, such as detection of food or danger evasion. Often in nature, these cues form perplexing spatial gradients and organisms need to accurately interpret the information and navigate accordingly via a process known as chemotaxis. Even single-cell organisms are capable to chemotax, and they do so using the biased-random walk strategy (klinokinesis). According to this strategy, cells increase or decrease turning rates depending on whether they move down or up a chemical gradient, accordingly[1]. To implement this strategy, cells adapt to absolute levels of the stimulus, an ability that also allows them to detect small changes in the environment[2,3]. The mechanisms that control this relatively simple strategy had been extensively studied to the molecular details[4,5].

Multicellular organisms, that harbor a neural system, use more sophisticated strategies when navigating based on chemical gradients. For example, chemotaxis of *C. elegans* nematodes is comprised of long periods of sinusoidal movement, termed 'runs', and intermittent turning events, where a bout of consecutive turns is known as a 'pirouette'[6–8]. In their seminal work, Shimomura et al.[8] demonstrated that *C. elegans* worms modulate the probability to perform a pirouette based on the sign of the first derivative of the sensed stimulus[8], similarly to the classical biased-random walk strategy observed in single-cell organisms. However, and unlike unicellular models, worms show a clear directional bias when exiting the pirouette: worms entering a pirouette following a run that was directed toward the target are more likely to exit the pirouette in the same goal-directed angle; Conversely, when entering a bout of turns following a run that was directed opposite to the target, then the exit angle is likely to be closer to 180°, thus reorienting the animals toward the target[8,9]. Later studies showed that in addition to modulating pirouette rates based on the sign of the first derivative, worms also take into account the magnitude of the derivative[9–11]. In addition, animals use a second navigation strategy in chemical gradients, termed klinotaxis[12–14]. In this strategy, animals continuously make smooth and gradual curvature corrections toward the chemical source, in a process termed in *C.elegans* 'weathervane'[13].

To support such complex navigation maneuvers, neural circuits perform various computations. These include adaptation and temporal integration of the sensed concentrations[7,15]; coding the magnitude of the change in the concentration[15–17], bilateral coding[18,19], and temporal coding[7,20]. To study coding principles and computations performed by individual neurons and circuits, it is useful to focus on animal models with a defined nervous system. In that respect, the *C. elegans* nematode offers a unique opportunity: it consists of a compact nervous system (302 neurons in total) for which a detailed wiring diagram is available[21]. Indeed, studies characterized worm chemotaxis behavior[8–10,22–24], as well as the neural response to a variety of different stimuli[25–27]; Furthermore, recent advanced experimental systems measure neural activity in freely behaving animals, allowing to infer the neural correlates of chemotaxis behavior[17,25,28,29].

Traditionally, chemosensory activity in *C. elegans* has been studied by presenting chemical cues in an on/off step-like manner, while simultaneously imaging activity from target neurons[18,25,26,30–32]. These step-like stimulations may approximate turbulent plumes, where signals are patchily distributed, and animals are exposed to cues that rapidly fluctuate in time and space[33–35]. However, animals, particularly small-size animals, are often found in limited and confined environments. For example, *C. elegans* worms are frequently recovered from rotting fruits[36], which constitute a secluded and turbulent-free environment, where abrupt changes in concentrations are presumably uncommon. In such settings, stable gradients may be formed due to diffusion from bacterial microenvironments or food deteriorating signals[37]. These gradients are expected to be smooth and continuous due to simple spatiotemporal diffusion processes. Under these conditions, animals are likely to exploit the fine features of the gradient, such as its first derivative and its delicate spatial and temporal structure. Indeed, numerous recent studies exposed new layers of neural dynamics in response to temporally changing stimuli[7,38–41].

The odorant diacetyl is one example for a signal secreted from bacteria in rotting fruits[42]. *C. elegans* worms strongly attract to this odorant, as it potentially indicates food sources. Diacetyl is sensed by two pairs of amphid sensory neurons in the worm—AWA and AWC[17,22,26]. Furthermore, AWA is the only neuron that expresses the diacetyl GPCR, ODR-10, which is required for chemotaxis toward low concentrations (<10 μM) of diacetyl[27]. Behaviorally, AWA activity has been shown to suppress turning events[39], while AWC activity is correlated with reversals[17]. In response to an abrupt increase in diacetyl levels, AWA activity transiently increases and eventually adapts to the new concentration, in a history-dependent manner[39]. In contrast, AWC responds to an off step in diacetyl[26]. Both AWA and AWC are connected to first-layer interneurons (e.g., AIY and AIA) that control worm navigation[39,43–45].

Here we study how individual neurons in *C. elegans* worms code smooth gradients, and how their neural dynamics dictate navigation strategy. Surprisingly, we find a previously uncharacterized mechanism, where AWA neurons pulse in response to smooth increasing gradients. Moreover, this pulsatile activity adapts to the magnitude of the gradient derivative, suggesting that navigating animals continuously seek steeper gradients, which correspond to trajectories better oriented toward the target. More fundamentally, we show that the orchestrated circuitry dynamics of AWA and AWC enables an efficient navigation strategy that outperforms the classical biased-random walk strategy.

## Results

**Smooth increasing gradients are coded by pulsatile activity**. To systematically study neural responses to continuously changing chemical gradients, we developed a microfluidics-based system that can present *C. elegans* nematodes with a wide range of smooth gradient shapes (Fig. 1a, see Supplementary Note 1 for details). We used this setup to study the response of the AWA neurons to a gradient of diacetyl. AWA neurons responded with a single pulse when presented with a 1.2 mM on-step of diacetyl (Fig. 1b, c). Similar single-pulse responses to a range of on-step levels of diacetyl were observed by others as well[39]. The increased activity decayed back, close to basal levels, after ~1 min, and remained at these low levels for the duration of the experiment (20 min), despite the fact that the stimulus remained constantly on. However, when presented with monotonous increasing linear gradients, AWA neurons constantly pulsed throughout the experiment (Fig. 1d, e, Supplementary Movie 1). Each pulse typically lasted for several tens of seconds and was characterized by a sharp increase followed by an exponential decay (Fig. 1f).

Notably, pulsatile characteristics (e.g., amplitude, frequency, and duration) significantly varied among animals, but each individual animal maintained a typical pulsatile signature ($p < 10^{-6}$, Fig. 1g, Supplementary Fig. 1). Furthermore, this pulsatile activity is cell autonomous as mutant strains defective in either neurotransmitter release[46] or neuropeptide secretion[47] exhibited vigorous pulsatile activity (Supplementary Fig. 2). While primarily cell autonomous, other neurons may be modulating this pulsatile activity, as the parameters depicting this activity slightly varied in the neurotransmitter and the neuropeptide defective animals (Supplementary Fig. 3).

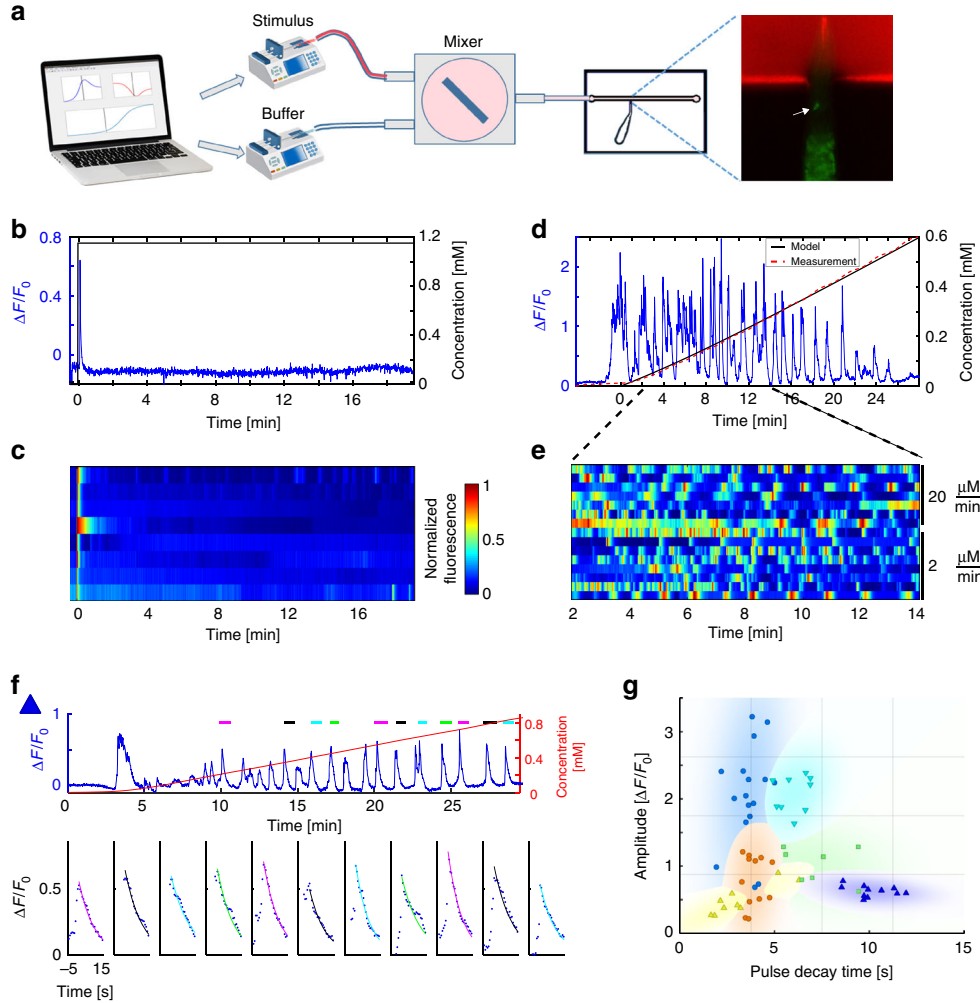

**Fig. 1** Smooth increasing gradients are coded by a pulsatile activity. **a** Illustration of the experimental setup. Two computer-controlled syringe pumps flow the stimulus (diacetyl with red dye, top) and the buffer (bottom) into a mixing chamber. The homogenous stirred mixture flows through a restrained worm inside a microfluidic device. GCaMP (green) and rhodamine dye (red) intensities are imaged. The right panel shows the restrained worm inside the chip, with the red dye in the flow channel. The white arrow points to the AWA neuron. **b** A step function of diacetyl (black) results in a single pulse. **c** AWA activity in response to a step function of diacetyl ($N = 8$). The first row depicts the dynamic response shown in **b**. **d** AWA shows pulsatile activity (blue) in response to a linear gradient of diacetyl. Shown is a sample trace of a single animal. The system outputs accurate smooth gradients, as evident by the excellent agreement between the measured (red) and the modeled (black) concentrations. **e** AWA pulsatile activity is observed for various linear gradients; top, response to a linear gradient with a slope of 20 μM/min ($N = 7$ animals); bottom, response to a 10-fold shallower linear gradient of 2 μM/min ($N = 8$ animals). The first row depicts the dynamic response shown in **d**. **f**, **g** Individuality in pulsatile responses. **f** Extracting pulse parameters from AWA activity (blue) in response to a linear gradient (red). Exemplary pulses are marked by a colored line on top. To each of the marked pulses, we fit an exponential decaying function in the form: $A \cdot e^{-\frac{t}{\tau}}$ (bottom, colored line), from which we extracted pulse amplitude and decay time. Pulses are ordered sequentially and the line color matches the line color above the activity trace. **g** Exponentially decaying pulses of six different worms (each denoted by a different color or shape) are plotted in the amplitude and decay time space. Pulses originating from the same animal were significantly more similar than pulses measured from other animals ($p < 10^{-6}$, bootstrap, see methods). The eleven pulses extracted from **f** are marked by blue triangles

**Pulsatile activity dictates behavioral outputs**. We next asked whether freely moving worms also exhibit pulsatile activity during chemotaxis, and if so, do these pulses carry a meaningful information that dictates navigational outputs. For this, we developed an automated tracking system that enables measurement of neural activity in animals that freely-navigate through chemical gradients (Fig. 2a, see methods for details). In agreement with the results obtained using the microfluidic device, activity of AWA neurons rose and decayed even when animals moved strictly toward the chemoattractant source (and hence experienced continuously increasing concentrations only, Fig. 2b–d, Supplementary Movie 2). This neuronal activity correlated with subsequent behavioral outputs: worms continued to move forward as long as the AWA neurons were active, and in most cases,

turned only after AWA activity considerably dropped (to 58% ±15 std from the maximal level, Fig. 2c–e). To further understand how AWA activity modulates worm behavior, we used Chrimson[39,48] to light-activate the AWA neuron. We found that in times that AWA was active, turns were significantly suppressed ($p < 10^{-10}$, X² test, Supplementary Fig. 4, and Supplementary Movie 3). This result is in agreement with Larch et al.[39], who additionally found that immediately after AWA activation, turning probability increased above baseline. Taken together, these results demonstrate that AWA activity dictates forward movement in times that AWA is rising, and a turn once its activity decays. Thus, pulsatile activity may induce turns even in animals that sense increasing gradients only, as they move toward the chemoattractant source. Indeed, high-throughput chemotaxis

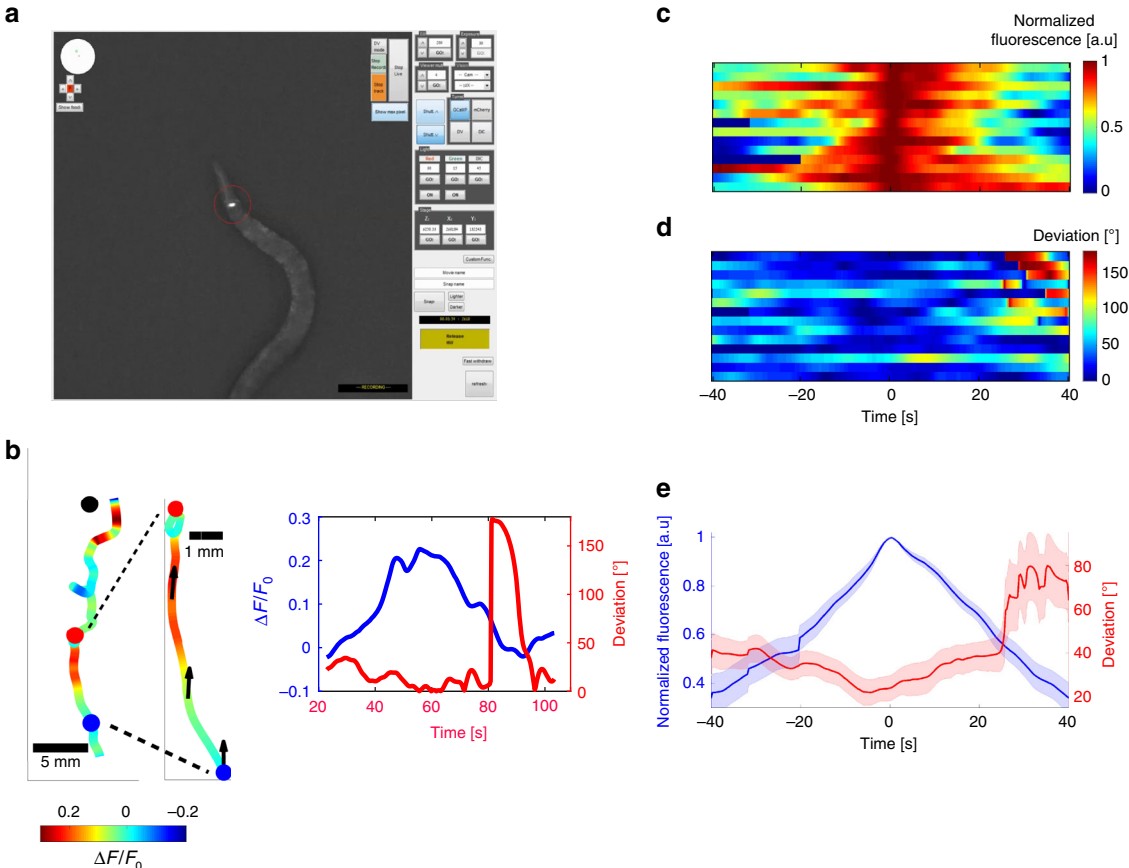

**Fig. 2** The pulsatile activity dictates behavioral outputs. **a** We developed an automated tracking system that enables tracking of worm movement while simultaneously measuring neural activity from a single neuron. Shown is a screenshot taken from the system as it tracked animal trajectory and AWA activity (red circle) during chemotaxis. **b** Left part is an example of a worm trajectory from its starting position to the target (diacetyl spot, black dot). The color code represents neural activity levels. In the middle, the trajectory zoom-in demonstrates a single pulse in which activity rose and dropped even when the animal was well directed toward the target (indicated by black arrows). This recapitulates the data observed using the microfluidic device, where activity increased and decreased despite continuously increasing gradients. Right graph extracts from the zoomed trajectory the fold-change activity in AWA (blue) and the angular deviation (red) between the velocity vector and the direct path to the target (as denoted by black arrows on the trajectory). Small deviations indicate that the worm is well oriented toward the target, and ~180° deviation indicates a movement in the opposite direction. An abrupt increase in deviation indicates a reversal event. This demonstrates that animals reverse after AWA crosses peak activity, during the decreasing phase of the pulse. **c** AWA activity measured from freely-chemotaxing worms ($N = 14$). The first pulse of AWA was aligned to the point of maximal activity. The first trace belongs to the worm plotted in b. **d** The angular deviation corresponding to the neural activation shown in **c**. All reversal events appeared after AWA crossed its peak activity, suggesting it is the decrease in AWA activity that promotes turns, and not vice versa. Maximal deviation is significantly higher in the second half of the pulse, during its decreasing phase (Wilcoxon rank-sum test, $p = 0.0035$). **e** Mean normalized intensity (blue) and angular deviation (red) of the 14 worms in **c**, **d**. Shaded colors mark the standard errors

assays demonstrated that animals may turn even when directed toward the target (Supplementary Fig. 5). These results suggest that pulsatile activity at the primary chemosensory neurons may directly dictate forward locomotion and turning events.

**Animals adapt to the magnitude of the first derivative**. The gradient formed by a single odorant source, in a non-fluctuating environment, is typically Gaussian, and animals navigating toward this source are likely to encounter a gradient with increasing first derivatives (Supplementary Fig. 6). Previous reports demonstrated that neural activity and behavior correlate with the first derivative of the stimulus[39,49]. As pulsatile activity is highly variable between animals (Fig. 1g), we analyzed neural responses of individual animals while exponentially increasing the gradient first derivative over time. We found that the pulsatile activity correlates with the first derivative of the gradient: the larger the first derivative, the higher is the amplitude ($r = 0.44$, $p < 10^{-4}$) and the shorter is the time interval between pulses ($r =$

$-0.31$, $p = 0.01$, Supplementary Fig. 7 and Supplementary Movie 4).

However, worms facing sigmoidal gradients fail to adhere to this principle. To test whether coding of the first derivative magnitude is time invariant, we designed our fine-controlled microfluidic system to generate a sigmoidal (namely, hyperbolic tangent) gradient (Fig. 3b), where the first derivative of the concentration monotonically increases to reach its maximum value exactly at the midpoint of the gradient, after which the first derivative values symmetrically decrease (Fig. 3c). This symmetry around the gradient midpoint suggests that pulsatile activity would also be symmetric around the midpoint. Surprisingly however, we found that this is not the case: instead, the activity was significantly lower during the period after the midpoint ($p < 0.002$, Wilcoxon signed-rank test, Fig. 3a, and Supplementary Movie 5). Thus, although worms encounter the same first derivative values before and after the gradient's maximal value, activity is significantly stronger in the first half of this function,

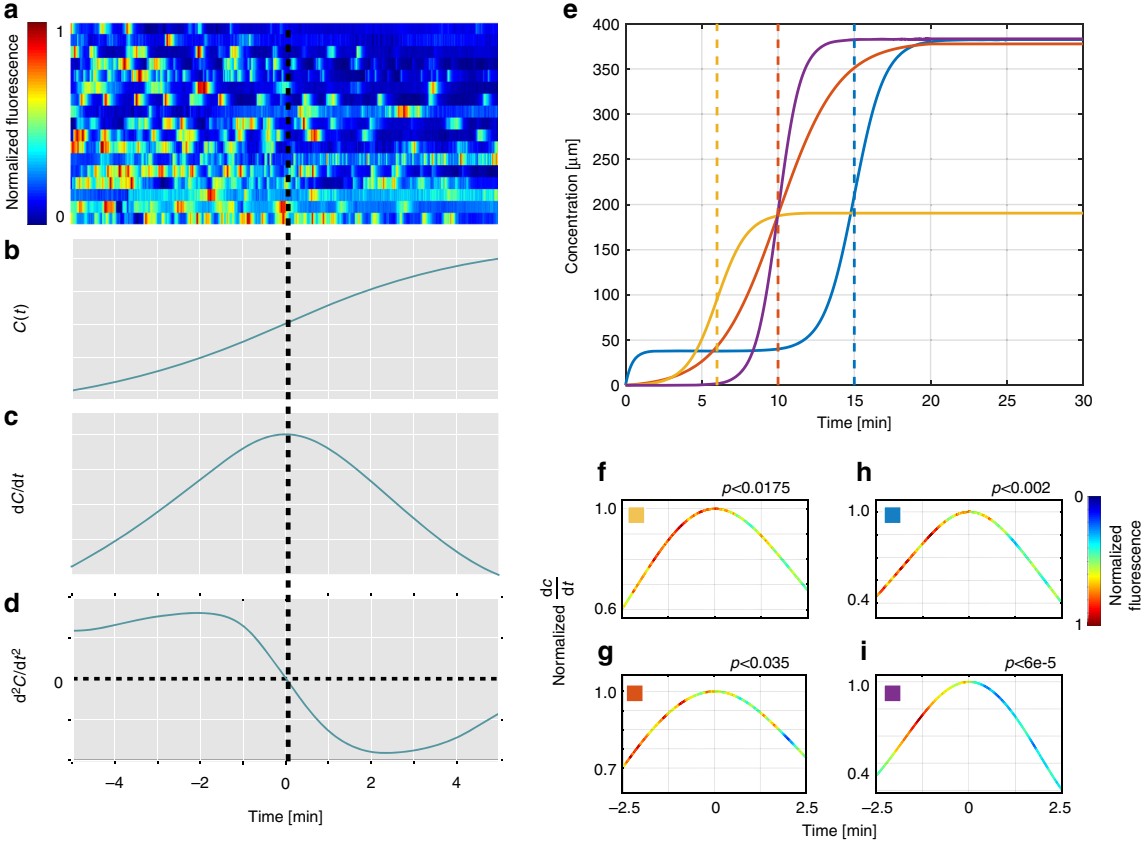

**Fig. 3** Worms adapt to the magnitude of the first derivative. **a** AWA neural responses in response to a sigmoidal gradient ($N = 17$). Responses are aligned to: **b** concentration (a hyperbolic tangent function); **c** first derivative, and **d** the second derivative of the concentration. Worms adapted to the levels of the first derivative, and thus, neural activity dropped asymmetrically after the first derivative peaked and the second derivative flipped signs from positive to negative. **e** Activity decrease after crossing the maximal first derivative point was due neither to the prolonged exposure to the stimulus nor to the stimulus concentration. To eliminate those possibilities, we shifted the stimulus function in time, and changed its absolute concentration. The blue curve is the sigmoid function shown in **a**–**d**. The red curve was shifted back by 5 min, such that the peak of the first derivative was reached after 10 min; the orange curve was shifted back by 9 min and reached absolute levels that were two-fold lower. The purple and the red curves reach the peak of the first derivative exactly at the same time; however, the gradient of the purple curve is two-fold steeper at the midpoint. **f**–**i** The mean neural activity for the sigmoid functions shown in **e**. The curve shows the gradient first derivative and the neural activity is color coded. In all cases, neural activity was significantly higher during the first half of the gradient, before the first derivative reached its maximal level (and where the second derivative was still positive) than during the second half, after the first derivative crossed its maximal level (and where the second derivative became negative); **f** $p < 0.02$, $N = 21$; **g** $p < 0.04$, $N = 12$; **h** $p < 0.002$, $N = 17$; **i** $p < 10^{-4}$, $N = 15$ (Wilcoxon signed-rank test)

suggesting that activity may actually adapt to the magnitude of the first derivative.

To verify that neural activity genuinely adapts to the first derivative, rather than simply adapting to the prolonged exposure to the stimulus, we analyzed activity while varying various parameters: the time to reach the midpoint, the absolute concentration values, and the maximal slope of the sigmoid at its midpoint (Fig. 3e). Remarkably, regardless of all these changes, AWA neurons activity significantly declined when crossing the midpoint, where the values of the first derivative began to decrease (Fig. 3f–i). Thus, the abrupt activity decline cannot be explained by the continuous exposure to the stimulus. Moreover, this activity cannot be explained by the mere first derivative of the concentration, nor by the fold change of the stimulus concentration (Supplementary Fig. 8). Together, these findings point to an intriguing principle, whereby neural activity adapts to the magnitude of the first derivative of the gradient. Interestingly, the midpoint of maximal first derivative also marks the position where the second derivative flips its sign from positive to negative (Fig. 3d). This raises the intriguing possibility that animals may actually sense the sign of the second derivative of the gradient (Supplementary Fig. 9a).

**A concerted neural dynamics underlies efficient navigation.** We next studied the dynamics of AWC^ON, an additional diacetyl-sensing neuron[26] in smooth gradients; to measure the concomitant activity of both AWA and AWC^ON, we generated a transgenic line expressing GCaMP in these two neuron types. In contrast to AWA neurons, it is the removal of diacetyl that elicits AWC^ON activity[26]. We therefore subjected the animals to periodic sinusoidal gradients of diacetyl (Fig. 4a). While AWA neurons exhibited the typical pulsatile activity, AWC^ON response was graded: it decayed during the increasing periods of the gradient, and was activated during decreasing periods of the gradient, in an anti-correlated manner such that it followed the first derivative of the gradient ($\rho = -0.74$, $p < 10^{-20}$, Fig. 4a–c, and Supplementary Fig. 9). Furthermore, while AWA pulsatile activity was highly variable among the individual animals, AWC^ON response was robust, deterministic, and highly correlated across all tested animals ($\rho = 0.83 \pm 0.1$, Fig. 4a–c, and Supplementary Fig. 9).

We next studied how the pulsatile activity propagates down the neural network to the AIY interneuron. In that respect, AIY is a key postsynaptic interneuron of AWA and AWC, whose activity correlates with forward movement, and its decreased activity

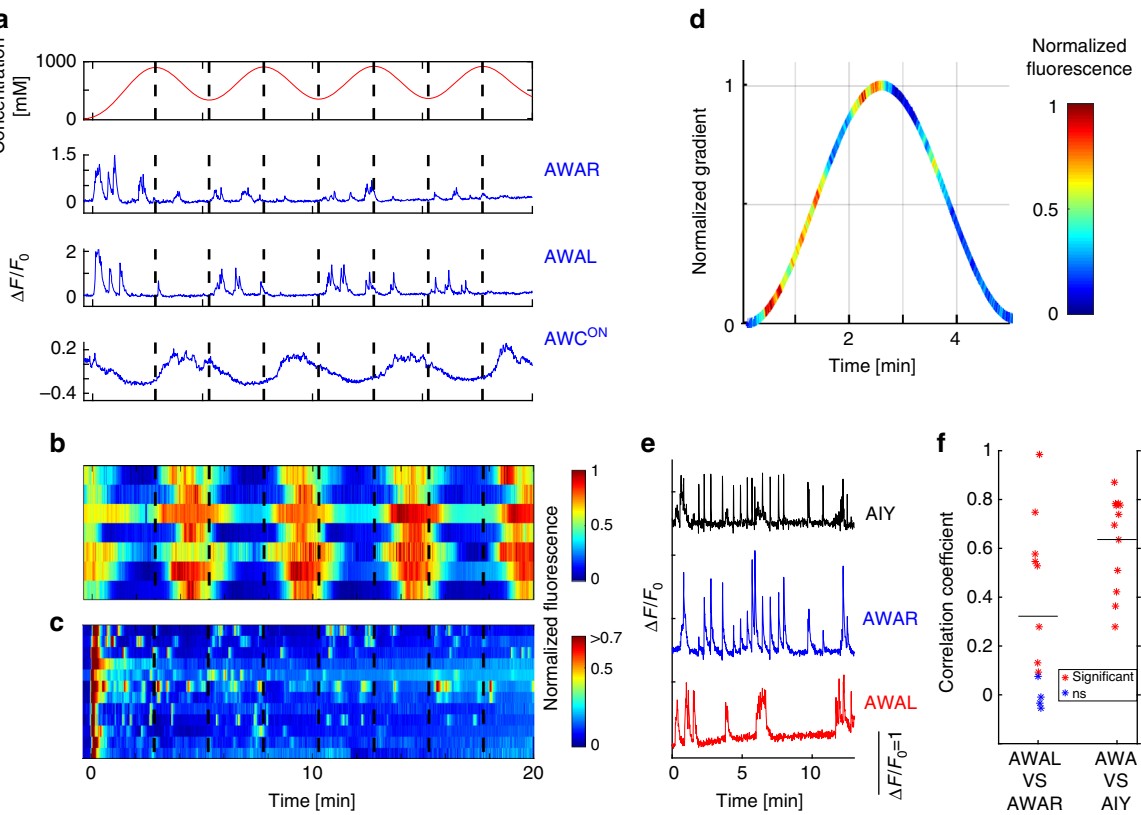

**Fig. 4** AWA and AIY pulse while AWC$^{ON}$ shows a robust graded response. **a** Neural activity of AWAR, AWAL and AWC$^{ON}$, as recorded in a single animal in response to a sinusoidal gradient (top). While AWA neurons stochastically pulsed, AWC$^{ON}$ showed a robust graded response that followed the gradual change in the gradient (for a more quantitative analysis see Supplementary Fig. 9). Dashed black lines mark the times where the gradient first derivative is zero. **b, c** Neural responses of AWC$^{ON}$ (**b**) and AWAR/L (**c**) in response to the same sinusoidal gradient as in **a**, $N = 7$. Top row in **b** and top two rows in **c** are the responses shown in **a**. AWC$^{ON}$ activity was significantly stronger in the decreasing parts of the gradient ($p < 0.0002$, Wilcoxon Signed-Rank test). **d** Mean response of the AWA neurons during a single period as calculated from all 14 AWA neurons over the second and third periods of the sine function. AWA activity was significantly stronger when the gradient was positive ($p < 0.009$, Wilcoxon signed-rank test). **e** An example of a neural activity from the two AIY neurons (black), AWAR (blue) and AWAL (red) in response to a sigmoidal gradient. Note that activity of the two AWA neurons was often asynchronous. AIY pulses were narrower and were synced with either one of the AWA neurons. **f** The pulsatile activity between the bilateral symmetric AWAR and AWAL neurons was weakly correlated ($\rho_{mean} = 0.32$). Conversely, the mean correlation in activity between the sum of the two AWA neurons and AIY was significantly higher ($\rho_{mean} = 0.64$, Wilcoxon rank-sum test, $p < 0.03$, $N = 12$ different animals). Black horizontal lines mark the mean correlations

mediates reversals[31,44,45]. For this, we generated a transgenic animal expressing GCaMP in both AWA and AIY neurons. When we imaged both neurons simultaneously, we found that the AIY neurons also pulsed, and these pulses highly correlated those observed in the AWA neurons ($\langle \rho \rangle = 0.64$, Fig. 4f, and Supplementary Fig. 10). However, AIY activity decayed significantly faster than that of AWA ($t_{1/2} = 1.4$ s vs 2.1 s, respectively, $p < 10^{-8}$ Wilcoxon rank-sum test), possibly explaining why worms perform turns at times when AWA activity is well above its basal level (Fig. 2c–e). Since AIY pulsatile activity was observed in the neurite extension, it was practically impossible to differentiate between the bilateral AIYR and AIYL activities; however, several visually distinguishable instances demonstrated that AWAR and AWAL activity correlated with their ipsilateral AIY partners (Supplementary Movie 6). Notably, although AWAR and AWAL neurons are electrically coupled[21], their activity is not well correlated; often, only one of the two neurons (either AWAR or AWAL) is active and pulsing, and in other cases both are active but pulse asynchronously (Fig. 4f, and Supplementary Fig. 10).

Taken together, when moving up a gradient, AWA and AIY pulsatile activity promotes forward movement, and the reduced

AWC$^{ON}$ activity negatively correlates with turning events. Conversely, when moving down the gradient, AWA activity is significantly lower ($p < 0.009$, Wilcoxon signed-rank test, Fig. 4d), thereby reducing prospects for forward movement, and AWC$^{ON}$ activity rises to promote turning events[17,31,44].

**First derivative adaptation, enables efficient navigation.** To understand the possible benefits of adapting to the gradient first derivative, we used simulations to compare the performance of two chemotaxis strategies that obey simple and general set of rules: (1) the well-established biased-random walk strategy which relies solely on the sign of the gradient first derivative[1,8,23], and (2) the same strategy, but in addition, the animals also sense the magnitude of the first derivative and adapt to it. Strikingly, we find that the latter strategy always outperforms the classical biased-random walk strategy, as animals using this strategy always reach their target faster, regardless of the values used for the different parameters (Wilcoxon signed-rank test, $p < 10^{-70}$, Fig. 5a–c and Supplementary Fig. 11). This is due to the fact that using the strategy that employs adaptation to the first derivative, animals continuously seek trajectories with larger first derivatives

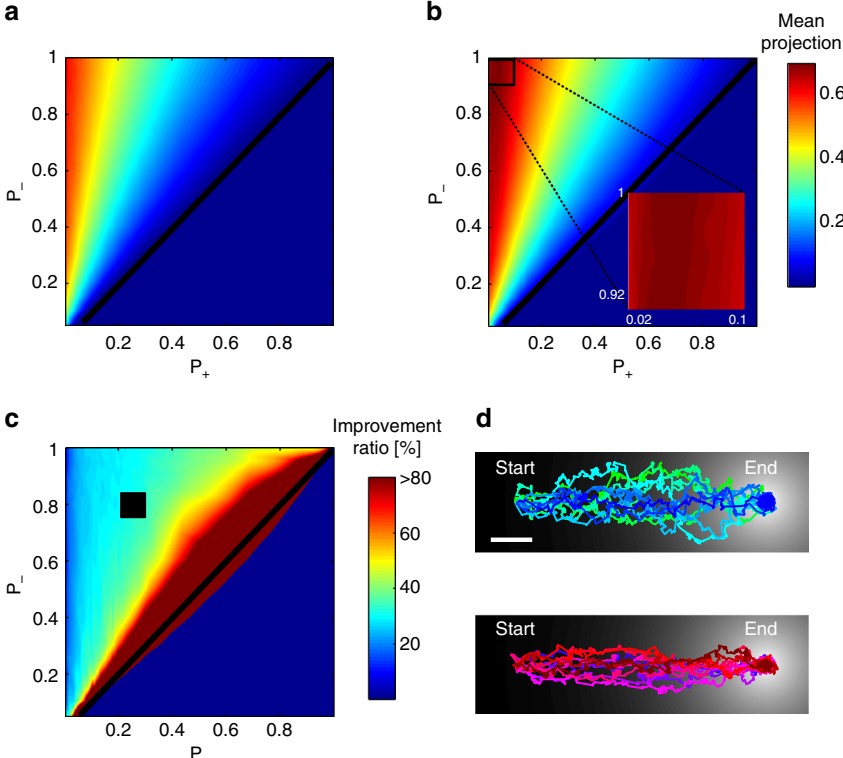

**Fig. 5** Implementing adaptation to the first derivative promotes efficient navigation **a**, **b** Chemotaxis performance when simulating: **a** the classical biased-random walk strategy, and **b** the strategy which incorporates adaptation to the first derivative. Axes denote turning probabilities when experiencing a positive gradient (x-axis, $P_+$) and a negative gradient (y-axis, $P_-$). Color marks chemotaxis score (performance), measured as the mean projection of the velocity vector on the optimal (direct) trajectory line (see methods for details). Black diagonal marks the $P_+ = P_-$ line. Zoomed-in box of the top-left square shows the maximal chemotaxis score. Interestingly, the maximal score is achieved for $P_+ > 0$, meaning that better performance is achieved when allowed to occasionally reorient, even if sensing continuous increasing gradients. **c** Fold-improvement in the performance between the strategy that incorporates adaptation to the first derivative (**b**), and the classical biased-random walk strategy (**a**). The fold-improvement is defined as 100·(first derivative adaptation score—classical biased-random walk score)/(classical biased-random walk score). Throughout the entire parameters' space, the strategy of adapting to the first derivative outperforms the classical biased-random walk strategy. When approaching $P_+ = P_-$ slope (black line), this ratio approaches infinity, because when implementing the classical strategy one can no longer navigate to the target. Black square is the area from which the tracks from **d** were taken. The heat maps are based on overall $5 \times 10^5$ simulated rounds, 800 for each $p_+$ and $p_-$ values. **d** Ten representative tracks of animals implementing the classical biased-random walk strategy (top), and the first derivative adaptation strategy (bottom). The Gaussian gradient used in the simulations is also depicted (black and white background). Scale bar marks a distance of 50 simulation steps. Note that tracks implementing adaptation to the first derivative are more directed, resulting in better (faster) chemotaxis performance

that are better directed toward the target (Fig. 5d). Importantly, the superior performance of the latter strategy is insensitive to the parameters chosen, as similar results are obtained for a wide range of parameters; for example, when varying the adaptation memory time, or when adding noise to the gradient sensed by the animal (Supplementary Figs. 12, 13).

Moreover, the simulations show that best performance is obtained when animals that adapt to the first derivative are also allowed to occasionally make turns, even when moving up the gradient (thus $P_+ > 0$ in the simulations, Fig. 5b inset, and Supplementary Fig. 11; see also Supplementary Note 2 for details). We find that this holds only in cases where the probability to turn due to negative gradients is sufficiently high (Fig. 5b inset). These simulation-based understandings may be projected onto the intricate dynamics observed for AWA and AWC$^{ON}$: AWA pulsatile activity, which also adapts to the first derivative, inflicts turns even when the animal is well oriented toward the target. AWC$^{ON}$ serves as a correcting mechanism, as its robust response to negative gradients promotes immediate turns[17,31].

## Discussion

In this work, we report a previously unknown navigation strategy for efficient chemotaxis. This strategy relies on the orchestrated dynamics of two chemosensory neurons: AWC$^{ON}$ and AWA. Remarkably, we found that the AWA chemosensory neurons code smooth gradients via pulsatile dynamics, and elucidated an intriguing principle, where the pulsatile activity adapts to the magnitude of the first derivative of the gradient.

Based on these findings, we propose the following model that underlies the efficient chemotaxis strategy (Fig. 6): When moving up the gradient, AWA pulses stochastically (Fig. 1d, e) thus inducing a forward run (Fig. 2a–c, and Supplementary Fig. 4). AWA also adapts to the first derivative, thus promoting a turn when the gradient begins to flatten, in search of a trajectory that is better oriented toward the target. However, turning based on the stochastic pulsatile activity of AWA comes at the risk of making erroneous turns that will draw the worm away from the source (Fig. 6). Therefore, this strategy may become advantageous only if combined with an efficient correcting mechanism. In that respect, this is exactly the role of AWC$^{ON}$, which responds robustly and

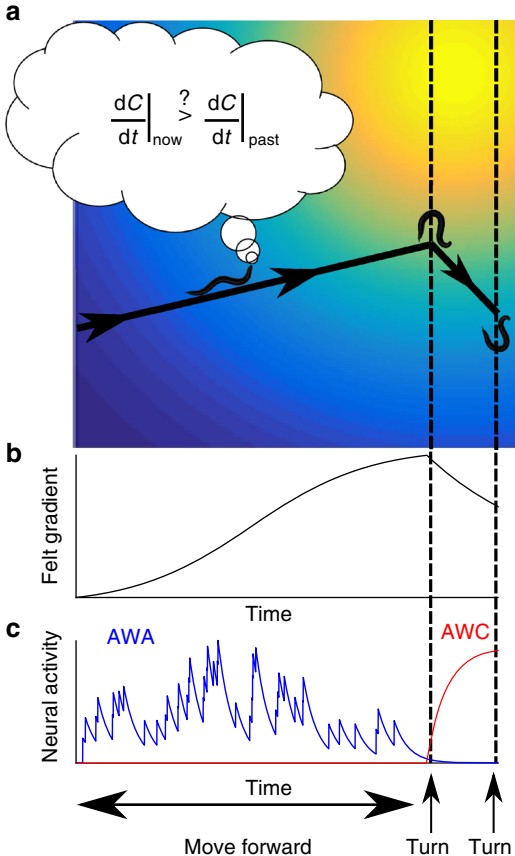

**Fig. 6** A mechanistic view of the neural codes for the efficient navigation strategy. **a** A trajectory of a navigating animal in a Gaussian spatial gradient of an attractant. Dashed lines mark turning events. **b** The temporal gradient sensed by the navigating worm in search for the attractant source. **c** AWA and AWC dynamics in response to the sensed gradient. AWA stochastic pulsatile activity controls the worm forward movement. AWA adapts to the first derivative of the gradient, so when the gradient begins to flatten (and although the first derivative is still positive), AWA activity decays, thereby promoting a turn. In case the turn steers the worm away from the target, the negative gradient is immediately sensed by the AWC neuron, which responds robustly and deterministically to promote a second turn. Thus, the orchestrated activity of AWA and AWC underlies the efficient navigation strategy in chemical gradients

deterministically (Fig. 4b) to promote turning events when facing decreasing concentrations[17,31]. Indeed, our simulations show that in the presence of a reliable corrective mechanism, occasional turns, even when navigating up the gradient, may become beneficial (Fig. 5b, inset). Moreover, a bout of turns (a pirouette) bears a corrective mechanism, as animals often exit a pirouette with an angle that is better oriented toward the target (when compared to the angle before the pirouette)[8,9].

Interestingly, studies in *C. elegans* worms demonstrated how an interplay between two sensory neurons may have functional roles in coding environmental cues[7,18]. For example, the anatomically homologues ASEL and ASER gustatory neurons sense NaCl, where ASEL is stimulated by increases in NaCl concentration, and ASER is stimulated by decreases in NaCl concentration. Behaviorally, ASEL activity prolongs forward locomotion whereas ASER promotes turning events[18]. In an analogous manner, ASH neurons respond to an increase in the repellent odorant 2-nonanone, leading to a bout of turns, while the AWB neurons respond to a decrease in odor concentration, leading to turn suppression[7]. Thus, orchestrated dynamics of two sensory

neurons may be a common design to efficiently integrate environmental signals before relaying the information to drive the appropriate behavioral outputs.

AWA neurons show two different modes of adaptation to external gradients of diacetyl. In the first mode, the neurons adapt to absolute levels of diacetyl, allowing them to remain sensitive to any changes in the concentration (Fig. 1b-c, and[39]). This type of adaptation is used by many organisms as part of the biased-random walk chemotaxis strategy[39,50,51]. In the present study, we revealed a novel type of adaptation: the neuron can also adapt to the magnitude of the gradient's first derivative (Fig. 3). Interestingly, drosophila larvae were also found to integrate past derivatives of the gradient to modulate future neural responses[15]. We show that this adaptation is beneficial as it allows worms to constantly search for trajectories with increasing first derivatives, therefore choosing trajectories better directed toward the target (Fig. 5). Furthermore, organisms often navigate in changing environments and first derivative values may vary by orders of magnitude. Similarly to classical adaptation processes that support sensitivity to a wide range of signal intensities[52], here, adaptation to the magnitude of the first derivative may allow animals to code a broad range of derivatives when seeking shorter paths toward the source. Indeed, to support a robust coding in face of a wide range of derivatives, this adaptation occurs on relatively long timescales, typically, several AWA pulses which correspond to dozens of seconds (Fig. 3). This longer timescale extends the single bout of forward locomotion (a 'run') and encompasses a longer chemotaxis period in which the worm learns through adaptation the expected first derivatives in its surrounding environment. Theoretically, adaptation to the first derivative on much shorter timescales may also be possible, in which case, animals will be actually calculating the second derivative of the gradient.

This efficient navigation strategy we report herein, where animals adapt to the first derivative of the gradient, joins other complex navigation strategies employed by multicellular organisms. These include a turn bias, where animals are likely to exit a pirouette better oriented toward the target[8,9], and klinotaxis, where animals make gradual curvature corrections toward the target[12–14].

Using simulations, we demonstrated that adaptation to the first derivative allows animals to perform better than the classical biased-random walk strategy (Fig. 5). Previous navigation studies also used simulations to provide a model that recapitulates the observed behavior. These type of simulations incorporated specific behavioral characteristics in order to reproduce the fine navigation features that were experimentally observed[7,13,15]. However, herein, we deliberately kept our simulations as general as possible as their sole purpose was to contrast between the two navigation strategies. Instead, we varied the various parameters of the model to show that regardless of their specific values, the strategy that employs the first derivative adaptation is always superior to the classical biased-random walk strategy.

We observed a large variability among individual worms in the AWA response to smooth gradients (Fig. 1g). Our findings (Fig. 2, and Supplementary Fig. 4), as well as those obtained by Larsch et al.[39], indicate that AWA activity dictates behavioral outputs (e.g., runs and turns). Thus, the high variability in AWA activity may underlie, and presumably explain, at least some of the extensive behavioral variability often observed in chemotaxis assays (see for example, chemotaxis assay reported in Itskovits et al.[9]).

An additional type of neural variability lies in the activity of the left-right, anatomically symmetric, AWAR and AWAL neurons. Interestingly, in some of the worms, either AWAR or AWAL responded to the gradients, while in other worms both neurons

responded. Furthermore, even in cases where both neurons responded, they often differed in their activation patterns (Fig. 4e, f, and Supplementary Fig 10). This differential dynamics may be particularly interesting in light of the neuroanatomical connections that these neurons make[21]. When analyzing the available connectome, both neurons directly connect (either by chemical synapses or via gap junctions) to the first layer of interneurons (e.g., AIY, AIA, and AIZ). However, unlike AWAL, AWAR is also synapsing onto deeper layer interneurons, namely, RIR, RIFR, and RIGR. This differential wiring may hint to variable outputs depending on the origin of the pulsing neuron (right or left). However, in our analyses, we could not systematically characterize a difference between the activation patterns of AWAR and AWAL.

In summary, here we report of an intriguing mechanism that underlies an efficient navigation strategy in chemical gradients. This mechanism is based on the principle that animals continuously seek increasing changes in the gradient by adapting to its first derivative. A handful of neurons suffice to implement this mechanism, suggesting that animals with higher brain systems may implement similar principles. Moreover, the underlying principle can be readily generalized and repurposed to support efficient navigation strategies utilized by other sensory modalities.

## Methods

**Strains used in this study**. AZS163 (gpa-6::GCaMP3, pha-1::PHA-1; lite-1;pha-1) was generated by crossing PS6390 with AZS43 (lite-1(ce314); pha-1(e2123)).

AZS164 (gpa-6::GCaMP3, pha-1::PHA-1; lite-1; pha-1; unc-13(s69)) was generated by crossing BC168 with AZS163.

AZS165 (gpa-6::GCaMP3, pha-1::PHA-1; lite-1; pha-1; unc-31(e928)) was generated by crossing AZS163 with AZS68 (unc-31(e928); pha-1(e2123)).

AZS281 (gpa-6::GCaMP3, str-2::GCaMP3,;str-2::dsRed; pha-1::PHA-1; pha-1; lite-1), where GCaMP is expressed in AWA neurons and in AWC^{ON}. We first generated AZS162 (str-2::GCaMP3, str-2::dsRed pha-1::PHA-1; pha-1; lite-1) and then crossed with AZS163.

AZS256 (gpa-6::GCaMP3, mod-1::GCaMP3; pha-1::PHA-1; pha-1; lite-1) was generated by injecting both constructs into the double mutant.

CX16573 (ky5662[odr-7::Chrimson::SL2::mCherry,elt-2::mCherry]; kyIs587 [gpa-6::G-CaMP2.2b, coel::dsRed])[39].

All strains were grown on NGM plates seeded with overnight culture of OP 50 according to Brenner[53]. L4 worms were picked aside the day before the experiment so that all experiments were performed on young adult animals at 20 °C.

**Microfluidic-based system for generating smooth gradients**. We developed a microfluidic-based system that allows generating a large variety of smooth temporal gradients (Fig. 1a). In this system, we control two syringe pumps (Chemyx fusion 400) using custom-made MATLAB code (Mathworks © Inc.). One pump holds a syringe with the chemical cue mixed with Rhodamine. The second pump holds a syringe filled with the diluting buffer (chemotaxis buffer (CTX)[8]). Importantly, Rhodamine alone did not elicit neural responses (Supplementary Fig. 14). Moreover, minute worm movements inside the microfluidic channel did not affect the observed neural responses as these displacements were uncorrelated to the pulsatile activity ($\bar{\rho} = 0.05$, Wilcoxon signed-rank test, $p = 0.15$, Supplementary Fig. 15). To verify accurate continuous flow of the gradient, and to avoid possible pressure buildup in the system, we used glass syringes (1000 series GASTIGHT, Hamilton).

Of note, diffusion, and possibly other fluid-flow processes in the tubing, causes a minute amount of the cue to arrive before its expected time based on calculation. This results in a neural response which may be observed up to 1 min ahead of its expected time. An example of such a case can be seen in Fig. 1d. Importantly, this is only a start-of-the-experiment effect, which does not affect the gradients to follow during the experiment. As soon as detectable levels of rhodamine enter the field of view, we can reliably quantitate them and accurately infer diacetyl concentrations at any given moment (Fig. 1a).

Both syringes flow through Tygon tubing (0.02' ID, Qosina Crop.) into a mixing chamber (of either 50 or 200 μL volume) with a small magnetic pole inside. The chamber is placed on a magnetic stirrer which ensures thorough mixing inside the chamber. The chamber output flows into a simple microfluidic device where the worm is restrained with its nose protruding into the flow channel (Fig. 1a). The microfluidic device is placed under the microscope for continuous imaging of the target neurons. Supplementary Note 1 includes a detailed description of the parameters used to generate each of the gradients presented in this study, along with the mathematical dynamical modeling of the system. In addition, the online information provides guidelines for the possible gradients one can generate using this system.

**Preparing worms and media**. Prior to imaging experiments, worms were placed on empty NGM plates (w/o OP 50) for a short starvation period (30–60 min). Following a wash in CTX, we inserted the worm into the microfluidic device designed with a short and wide flowing channel (L = 5 mm, W = 0.5 mm, h = 35 μm) to flow the gradients through the tip of the nose of a constrained worm. The wide channel lowers flow resistance and thereby increases gradient accuracy.

We used two syringes as the input to the mixing chamber, a 'Buffer' syringe and a 'Stimulus' syringe (Fig. 1a). The 'Buffer' syringe contained CTX buffer with 0.12 μM diacetyl. The purpose of this initial basal concentration in the buffer syringe was to increase accuracy of the flowing gradients by reducing possible noise and variability due to minute flow fluctuations.

The 'Stimulus' syringe contained diacetyl (1.15 or 0.115 mM) diluted in the CTX buffer. To verify the accuracy of stimulus gradients we added to the 'Stimulus' syringe a Rhodamine dye (0.2–1 μM). Sequential imaging in the red channel (for Rhodamine) and the green channel (for GCaMP signal) provided high-temporal resolution measurements of the gradient throughout the experiment.

To further increase measurement accuracy we reduced the effect of worm movement by adding 10 mM Levamisole (Sigma, CAS Number: 16595-80-5), similarly to previous reports[54]. Importantly, the addition of levamisole did not affect the pulsatile activity as similar results were obtained when using unanesthetized worms (Supplementary Fig. 16 and Supplementary Movie 7).

**Imaging single neurons**. Imaging one of the pair of AWA neurons was done using an Olympus IX-83 inverted microscope equipped with a Photometrics EMCCD camera and a 40× magnification (0.95 NA) Olympus objective. A dual band filter (Chroma 59012) and a 2-leds illumination source (X-cite, Lumen Dynamics) were used to allow fast iterative imaging of both green and red channels sequentially. Hardware was controlled using Micro-Manager[55]. AWA activation (green) and the Rhodamine concentration (red) were each imaged at a rate of 1.4 frames/s. We then developed in-house MATLAB scripts to analyze the movies and to extract neural activity. Notably, imaging at 1.4 Hz was indeed sufficient to reliably capture the pulsatile calcium dynamics (Supplementary Fig. 17).

**Imaging several neurons simultaneously**. Imaging of several neurons simultaneously was done using a Nikon AR1 + fast-scanning confocal system controlled by the Nikon NIS-elements software. We used a water-immersed 40× Nikon objective (1.15 NA) for imaging at a frame rate of 1.4 volumes/s. Pinhole opening was 1.2 Airy units and z slice jumps were ~0.7 μm. We then developed in-house MATLAB scripts to analyze the movies and to extract neural activity.

**Imaging freely behaving worms during chemotaxis**. For these experiments, we developed a software package based on the Micromanager[55] software suite for tracking and fluorescence imaging using a commercially available microscope setup. The code utilizes a motorized stage and a light source to track the worm while exciting and imaging its calcium sensor in 10× magnification. The code for this system, together with a detailed description of the entire system, can be found in our lab's github repository: https://github.com/zaslab/ FreelyMovingNeuronTracker. In this study, we used an Olympus IX-83 inverted microscope, with a 10× UPLASAPO objective, Lumen-Dynamics' X-Cite light source, Prior H117 motorized stage and Photometrics Evolve 512 camera.

We always assayed young adult animals following a short starvation period (30–60 min). A single worm was placed in a 5 μL CTX drop on an NGM agar plate at a distance of 30 mm from a 1 μL drop of diacetyl (1.2 Molar). To prevent external perturbations to the gradient, we concealed the experimental arena: we first placed 1 mm PDMS spacers on two opposing edges of the agar arena, along the formed gradient, and then placed on them a 43 × 50 mm glass coverslip. The coverslip did not come in contact with the agar but was 1 mm above it and the imaged worm. Given the diffusion constant of diacetyl is $D \approx 9 \, \text{mm}^2/\text{s}$ (calculated based on its molecular mass), it should take roughly $t = \frac{L^2}{D} = 30 \, \text{s}$ for the gradient to stabilize once the coverslip is placed. The worm was kept in the 5 μL droplet for approximately 3 min before the droplet evaporated, giving the diacetyl gradient enough time to stabilize.

Once the CTX drop evaporated and the worm emerged out of it, we started imaging (frame rate of 2 Hz). We then analyzed the movies using custom-made MATLAB software to extract neuron activity together with the worm position in relation to the chemical source. For accurate determination of worm trajectories, and to compensate for the wide-angle head swings during movement, we smoothed worm tracks (Supplementary Fig. 18) with a smoothing spline, that uses a least-squares approach with penalization for roughness[56,57].

**Analyses of high-throughput behavioral chemotaxis data**. We used previously published data[9] to analyze the turning rate of worms given their bearing in relation to the chemical source. In each of those experiments, approximately 100 worms were placed on one vertex of equilateral triangle with edge lengths of 4 cm, while diacetyl and dilution buffer were placed on the other two vertices. Tracking was done using the Multi Animal Tracker software suite[9].

**Light-activating AWA neurons in freely moving animals**. Worms expressing the Chrimson channel in the AWA neurons[39] were picked at the L4 larval stage and

separated into two groups. One group was picked into an NGM plate supplemented with 1 mM all *trans*-retinal (ATR, 100 mM stock was diluted 1:100 into *E. coli* OP50 prior to plate seeding). The second control group was picked to a NGM plate seeded with *E. coli* OP50 only. The following day, worms were randomly picked from either plate and subjected to three trials of behavioral analyses. During each trial we waited until the worm started a run and then exposed it for 10 s of green light (wavelength 545 nm, Bandwidth 25 nm, Intensity 14 mW/cm$^2$). During these 10 s, we inspected whether the worm performed a reversal. Interval between consecutive trials for the same worm was at least 15 s. Importantly, the experimenter was blind to the worm's group.

**Simulating chemotaxis performance**. We contrasted the performance of two chemotaxis strategies: The first obeys the sign of the first derivative only, and hence follows the classical biased-random walk strategy. The second strategy implements on top of the first strategy the ability to adapt to the first derivative of the gradient. These simulations were intended to examine the possible benefits that arise from adapting to the magnitude of the experienced first derivative, rather than simulating a fully-detailed model in attempt to fit the experimental observations. We therefore simplified our model as much as possible, and included only the parameters necessary to contrast between the two strategies. Supplementary Note 2 provides a detailed description of the simulation, including an analytical solution for the case of linear gradients.

**Variability and individuality in neural responses**. To analyze the variability of the pulsatile responses, a total of 92 discrete pulses were compiled from 10 different worms responding to a linear gradient (Fig. 1g depicts responses from 6 of the worms). To test whether each worm is characterized by significantly different pulse properties, we first calculated the standard deviations of different pulse parameters (namely amplitude, and pulse decay time) for each worm. We then shuffled all 92 pulses between the 10 worms, thus assigning each worm a random set of pulses, but each worm consisted with the same number of pulses it originally had. For this random set, we calculated the mean standard deviations for each of the pulse parameters and compared it to the mean standard deviations obtained for the original data. The results of this bootstrap analysis showed that the standard deviations of the random shuffles ($N = 10^6$ in total) are significantly higher than those of the original data ($p \leq 10e-6$). This analysis demonstrates that each worm responds with a characteristic pulsatile activity (which may suggest worm individuality) that is significantly different from pulses observed in other worms (population variability). Similar analysis was used to compare pulses amplitude and peak to peak time ($p \leq 10e-6$, Supplementary Fig. 1).

**Plotting neural activity**. Raster plots of neural activity are presented as heat maps (Figs. 1c, e, 2c, 3a, 4b, c, Supplementary Figs. 2a, b, 8a, 16). The values were normalized per each neuron (row) to range between 0–1:
$$[val - \min(val)]/[\max(val) - \min(val)]$$

**First derivative adaptation**. Neural activity within each worm was first normalized as described above. We then calculated the mean activity in each worm during the 2.5 min before and after the point of maximal first derivative. A non-parametric Wilcoxon signed rank test was used to compare the mean values of these two time periods (results are shown in Fig. 3).

**Pulse analyses**. To extract the parameters of individual pulses, (Figs. 1f, g, 4e, Supplementary Figs. 1, 3, 7, 16), we marked them manually. When pulse amplitude normalization was required (Supplementary Fig. 7), it was done for each pulse in respect to the other pulses observed in the same worm according to
normalized amplitude = $[amplitude - \min(amplitude)]/\max(amplitude)$Peak to peak time was similarly normalized.

**Code availability**. The code for imaging freely moving animals as well as the code for the simulations can be found in the github repository: https://github.com/zaslab/. Any additional data information is available upon request.

**Data availability**. The mean fluorescence values of the AWA neuron and the measured gradients throughout the experiments are available under the Open Science Framework.

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

## Acknowledgements

The authors thank Cori Bargmann for providing strains and reagents. The authors also thank Sagiv Shifman and Ami Citri for valuable comments on the first drafts of this manuscript. Some strains were provided by the CGC, which is funded by NIH Office of Research Infrastructure Programs (P40 OD010440). The project was funded by ERC (336803), ICORE, and ISF (1259/13, 1300/17) to AZ. EI and RR are also supported by the Jerusalem Brain Center.

## Author contributions

E.I., R.R., and A.Z. conceived the research. E.I. and R.R. performed all the experiments and analyzed the data. E.I., R.R., and A.Z. interpreted the results and wrote the paper. A.K. established the freely-behaving worm tracking system.

## Additional information

**Competing interests:** The authors declare no competing interests.

