## [Peer Review File · Nature Communications]

Reviewers' comments:

Reviewer #1 (Remarks to the Author):

Itskovits et al use the nematode *C. elegans* to probe the correlations between sensory neural activity dynamics and animal behavior. They present gradients of odor stimuli to the nose and monitor AWA, AWC and AIY neuronal activity patterns. They then use these data to generate a mathematical model that is predictive of directed locomotion. While these results are potentially interesting, I do have some concerns that should be addressed before publication. These are listed in no particular order.

Major

1. Do any of neurons (AWA, AIY and AWC) respond to rhodamine? Such a response would confound all of the measurements.
2. In most of the supplementary videos, there is significant movement. These motion artifacts (which should match the strength of the odor) are likely to affect the calcium responses.
3. Fig 1B: Max concentration used in step-function is 1.2 mM. Fig 1D: It is a linear function, max at 0.6 mM. Fig 3B: A sigmoid with a max of ~190 μ M or ~375 μ M. Fig 4A: Sinusoid with max close to 1 mM. Why are all these maxes different? Or, do matching them obscure the results presented herein?
4. Fig 1G: AWA in individual worms have unique pulsatile activity. Is this behaviorally meaningful? Optogenetics would be an ideal experiment to test this idea, but at least the mechanisms could be discussed.
5. Fig 3B: were there any attempts to increase the concentration at the sigmoid midpoint, or the length of time to achieve it? Furthermore, the authors modulate 1) time and 2) concentration. Time and concentration are both changed in the orange sigmoid, however, and it does not seem like the authors had a stimulus where only the time was changed. it might be interesting to do that experiment by using the same concentration midpoint of the blue (or red) curve.
6. Fig 4J: It is unclear to me that faster chemotaxis is better chemotaxis, since it decreases roaming and, hence, the ability to look for better food sources. Perhaps using shallower gradients would be a better test.
7. The differential pulsing between AWAR and AWAL neurons is very interesting. I would recommend discussing these data in the context of the known neuroanatomical connections with downstream neurons.
8. Fig S10: The authors were able to measure calcium transients in moving animals. Are there significant differences between these responses to ones obtained from animals that are anesthetized?
9. Supp Fig S2: Do pulsatile characteristics change when using the mutant strains defective in neurotransmitter release or neuropeptide secretion?
10. Simulation: $M = 30$ memory steps. What happens if fewer are used? How robust is the novel chemotactic strategy to changes in this parameter? (For instance, the authors changed M to 50, and find similar results, but no other values seem to be tested)

Minor

1. Supp Fig S3: presumably, the worm strains used in Ref. 8, which were used to make this figure, are the same reported in the rest of this article. They should also be in similar environments.
2. Fig 3B: it looks like the orange curve was shifted back by 9 minutes, not 10 minutes.
3. The authors imaged neurons in the microfluidic device at a rate of 1.4 frames/sec. This may be too slow to reasonably capture the activity of the studied neurons, though it does seem that the dynamics they saw at this imaging rate were slow enough to be mathematically described.

Reviewer #2 (Remarks to the Author):

Report:

First of all, my apologies for the delay in submitting my review.

Overall, I find this manuscript potentially interesting and intriguing, but for the reasons detailed below, I have the feeling it is an incomplete and, to some extent, a rushed piece of work. Also, in my opinion many important details are omitted and some specific facts are generously generalized. I thus encourage the authors to do MAJOR REVISIONS on the manuscript so as to be a solid, clear, and comprehensive research piece.

Title and abstract

Regarding the title, it reads too pompous: rather than announcing the results of a specific piece of research, it reads like the title of a classic book on the topic. What are those "principles"? The authors could write down specifically what they mean. What is that "neural coding"? One can be concrete and precise without losing generality. And, "for efficient navigation": well, it does not hurt to say for what organism, as the manuscript does not present results for many species. In sum, I think it is fair to the readers to say precisely what was found without making it sound more (nor less) than what it is.

Regarding the abstract, it seems to start with the usual "mantra" of "navigation is central to survival", and that is fine, but perhaps it is too much to say that "little is known about how animals navigate based on chemical cues". Perhaps there is still a lot to learn, but certainly a lot is known about chemotaxis, at many levels (behaviorally, molecularly, circuit-wise, etc) and in many organisms (bacteria, worms, larvae, and rodents).

The abstract insists on a "previously unknown mechanism" and a "novel principle". I think the authors could make a more in depth scholarly effort to mention previous results, both at the behavioral and coding levels, where very similar mechanisms have been found (ie. Schulze et al eLIFE on larval coding of temporal olfactory traces, , tons of work by the Bargmann lab, sophisticated analysis on worm navigation by Samuel lab, Lockery, etc, the Iino paper on weathervaning, also in the larva by Gomez-Marin and Louis, and even in rodents by Bhalla amongst others).

The abstract ends proposing that such mechanism is generalizable to other sensory modalities. Is it a rhetoric statement to try to increase generality of the findings, or is there a specific rational or evidence to claim that?

Main text

As mentioned for the abstract, the authors perhaps underestimate in the first paragraph what is

known about the neural and behavioral basis of chemotaxis.

In the second paragraph, could the authors specify what information step-functions carry, beyond saying they "carry little".

In the third paragraphs, they authors say "In agreement with previous reports," But there is no citation.

Also, I found the references regarding what is known about neural responses to temporally changing concentrations very scarce, poor and not making justice to the great progresses in the last 5 years or so, including previous pioneering work (in worms, but also in the larva, in locusts, and in rodents).

The development of the tracking system to enable measurement of neural activity in animals that freely-navigate through chemical gradients that they authors report is difficult to assess with respect to existing similar methods. Please provide more details and also discuss to what extent what is presented is novel or different. If it is not, that's fine too, but it must be specified.

Can the authors justify and specify the figures of the "high-throughput chemotaxis assays" they mention.

We read that "Naturally-formed chemical gradients are typically Gaussian", but in many cases turbulence may disturb those gradients, or simply their own diffusion will make them change in time. None of these possibilities seem to be contemplated. Also, it would be nice to have a reference or some arguments to better justify that worms indeed often encounter quasi-steady Gaussian chemical gradients. Note that then the authors concentrate on sigmoidal functions.

The "novel principle" their "findings point to" is certainly not novel beyond the worm. For instance, Schulze et al eLIFE 2015 explored a wide range of temporal gradients and showed, empirically and by means of analytic modeling and numerical simulations, that a single olfactory sensory neuron can adapt to changed in concentration, differentiate and integrate them, which, given the wide ranges of sensory protocols used in that study, and the very properties of their model, clearly showed that fly larvae detect and use information beyond the first temporal derivative. None of this is mentioned by the authors, nor regarding previous worm data.

Could the authors clarify why the so called "pulsative activity" is not simply noise? I do not see from their plots or their movies, how one would be confident that such signals is faithfully used by the worm to navigate. I am sorry if that was obvious from the presented results, but it was not obvious to me.

In Figure 1, could the authors clarify what they mean at the end of the figure legend by "as if each animal maintains a typical individuality in pulsatile response". If that is the case, I find this pretty interesting.

Figure 2, where it is claimed that "the pulsative activity dictates behavioral outputs", one can see an example of a worm trajectory plus its neural activity, which is said to recapitulate the data observed in the microfluidic device, but I do not see how faithful and extrapolable that is to claim that the pulsative activity dictates navigation. In what sense?

In Figure 3, I wonder why the temporal scales are so large, namely, of the order of minutes. Plus, these are mainly representative/illustrative plots, but no statistical claims are presented.

Similarly, in figure 4, we see some recapitulation of results in panel (e) for the correlation coefficient, but all the rest is essentially illustrative plots. Moreover, could the authors better specify what they mean by "orchestrated neural dynamics"?

The simulations on chemotaxis performance are interesting but somewhat limited. Could the authors show the effect of noise in the integration, and also the kernels that do so, and the typical timescales necessary and plausible in the past?

Also, very little is discussed about the AWA neurons, and its relation to previous works and situation in known circuits.

Supplementary Information

In the Supplementary Online Information I found just hints of crucial issues related to materials and methods.

First, in section 2, could the authors clarify if they really mean that the "developed microfluidic-based system allows generated ANY DESIRED smooth temporal function of a gradient", or, namely, to be quantitative about what is meant by "smooth".

Are any of the terms in the "Modeling system dynamics for generating temporal gradients" based on laws for fluid dynamics and, if so, could the authors specify which ones and why.

I am no expert in "imaging neural activity", and so I cannot comment in detail, but I presume some other experienced reviewer may want to know more about the details of it.

Regarding section (4), Imaging freely behavior worms during chemotaxis, I have several questions. To prevent external perturbations of the gradient, the authors use a coverslip above the imaged worm. From experience, I know that simply closing the cover slip can cause turbulence and therefore disturb the odor gradients. Could the authors comment on that and also provide quantitative evidence that the gradients are stable.

The freely-moving worm is imaged at 2Hz, which I think is not quite enough. Given the typical timescales of worm locomotion (body bending) and head movements, it has been standard for many years now to monitor at least at 15Hz, if not faster. And this is no big deal nowadays. Having 2 frames per second may miss much of the postural changes of the worm, with which it implements locomotion (including the "wide-angle head swings during movement" that the authors mention), and thus navigation, as well as miss the fast timescales related to the motor-sensory loops and perceptual dynamics from the perspective of the animal.

The custom-made worm tracker lacks all details. Please explain something about it. Similarly, the Matlab scripts used to control the tracker, track the worm and extract neuron activity should be made available in established repositories (ie. sourceforge or github) and/or included as supplementary material. Otherwise, it is a black box.

Regarding data analysis on the behavioral side, the only specification we read in the supplementary material is "we smoothed worm tracks". This is not enough. What filters? What time windows, etc, etc?

I found Point (5), the Simulating chemotaxis performance, interesting and valuable. Yet, again, the authors seem to think that details are irrelevant. For instance, given the Gaussian gradient they simulate, what were its mean and variance? And, did it evolve in time, as physical diffusion may

imply?

In the simulations, time goes as $t=T+1$, but can the authors specify what is the dt , and whether timescales matter with respect to real physical and biological units of the worm?

Note that using the sign of the temporal changes in concentration assumes some sort of normalization of concentration baseline. The authors could comment on previous evidence for that.

If I recall correctly, EColi does reorient randomly in any direction (and in 3D). As for worms, this may not be so (please double-check classical work by Shimomura and Lockery). This is relevant since in the simulation the authors "chose the new direction randomly from a uniform distribution".

It is very important to discuss further the role and values of the "memory length" M (it is only mentioned later in the text that $M=30$ is used, and that they "observe the same general behavior" for $M=50$, but what does that mean?), cause it may point to the integration properties of the worm (see for instance the discussion on the explicit quantitative model for Transient Normalization, including adaptation, normalization and differentiation in the *Drosophila* larva by Schulze et al. eLIFE).

Also, note that all physical and biological variables (ie. worm speed, distance to source, gaussian decay, time, etc) are [au], which means arbitrary units, but to have some plausibility, the authors should try to map them into the well-known typical scales of the worm.

The authors, when discussing their results on the projection score say that "we surprisingly find... meaning it is beneficial to occasionally reorient even when moving up the gradient". Well, that is not surprising at all, isn't it? If the animal moving up the gradient, it can still increase its navigation efficiency by aligning better to the local gradient. This has been shown clearly, for instance, in *Drosophila* larvae (Gomez-Marin et al, Nature Communications 2011), and it makes sense for any organisms that would want to improve its chemotaxis.

Also I find missing in the discussion the more subtle chemotaxis phenomenology known for the worm, such as the Iino paper on weathervaning (and its counterpart in the larva by Gomez-Marin and Louis in 2014).

Finally, from the simulations, the authors find out that including information of the first derivative improves the classical biased-random walk strategy. And this is what they call the "principle". If so, the authors should carefully review and report the literature, in worms and other organisms, to report where such beyond-random-walk strategies have been discovered or discussed empirically, numerically and theoretically.

Finally, and insisting on a minimum of ecological relevance, the authors say that a 2D exponential decay gradient (which is a Gaussian one, actually, rather than an exponential) "may better resemble genuine gradients animals are likely to encounter in nature". Can the authors provide some references to what is known in worms regarding such gradients?

Finally, in the section (7) Data analysis, could the authors give more details as to the relevance of bootstrapping from 10 worms data to 1 million random shuffles?

Therefore, I encourage the authors to take these suggestions into consideration in order to strengthen the content, clarity and significance of their work.

In order to increase transparency in the delicate and important process of reviewing our peers' work, I always disclose my name: Alex Gomez-Marin.

Reviewer #3 (Remarks to the Author):

Itskovits et al., have used *C. elegans* as a model system to understand the mechanism of efficient navigation in chemical gradients. To do this they have analyzed their response to external chemical stimulation. They have revealed that two chemosensory neurons, AWA and AWCON, have distinct responses, with AWA showing pulsatile responses in shallow odorant gradients and AWCON showing robust responses. Based on those experimental results, they use computational modeling to show how these two distinct functional chemosensory responses empower a more efficient navigation strategy than the classical biased random walk strategy. The manuscript is well-written and easy to read, and I think it is of high significance and appropriate for Nature Communications. A few experimental concerns are listed below.

The authors show (Fig 1d) clearly different responses to a step change of diacetyl vs a gradient from 0 to 0.6mM. It would be interesting to know what leads to this discontinuity. Have the authors tried much shallower gradients to see what the threshold for pulsatile responses is?

Also, in Figure 1d there appears to be a response in AWA before there is a change in diacetyl concentration. Is this a misprint, or is there an explanation for this apparent anticipatory response?

For Figure 1e, it was not clear from the text whether authors believe there is a difference between the responses observed in the $10e^{-8}$ to $10e^{-4}$ gradient vs the $10e^{-8}$ to $10e^{-5}$ gradient. If they believe there is a difference, they might want to plot out the data in a way that shows this more convincingly. In any case some clarification would be helpful.

In Figure 2a, is the software used for freely-moving imaging available for other researchers? It seems like a useful tool so this might be something to consider.

The authors use a t-test to compare imaging results (e.g. third sentence, para 9). It might be worth also running a non-parametric test as the results may not be Gaussian in distribution.

In Figure 3 the authors make the interesting claim that the neurons sense the first derivative of the chemical gradient rather than adapting to the prolonged exposure to the stimulus. However, it is possible that the first derivative only indicates the increment of concentration increase. To see if it is truly the first derivative that is being sensed, the authors could conduct similar experiments with a sigmoidal decreasing concentration of odorant.

A point-by-point letter response to reviewers' comments

We wish to thank the reviewers whose comments aided us to significantly improve the manuscript. Below, please find our detailed point-by-point response in which we have fully addressed all of the concerns. Specifically, we have now added new experimental conditions, controls, thorough analyses, and simulations, all further support the conclusions reported in this study. In addition, we now provide a detailed description of the relevant work in the field, and better explain the novelty of our study in light of what was previously known.

Reviewers' comments:

Reviewer #1 (Remarks to the Author):

Itskovits et al use the nematode *C. elegans* to probe the correlations between sensory neural activity dynamics and animal behavior. They present gradients of odor stimuli to the nose and monitor AWA, AWC and AIY neuronal activity patterns. They then use these data to generate a mathematical model that is predictive of directed locomotion. While these results are potentially interesting, I do have some concerns that should be addressed before publication. These are listed in no particular order.

Major

1. Do any of neurons (AWA, AIY and AWC) respond to rhodamine? Such a response would confound all of the measurements.

This is indeed an important control that we have now added to the revised manuscript. We measured the response of each of these neuron types (AWA, AIY and AWC) to a sinusoidal gradient of rhodamine. We chose to use the sinusoidal gradient to test possible responses to both increasing and decreasing gradients. The results of this control are shown in the figure below which now appears as a new supplementary figure S14. Importantly, we found that none of the neurons responded to rhodamine gradients. As is evident, AWA and AWC are active only when diacetyl is present in the experiment, and they are not active (flat lines) when diacetyl is absent and the stimulus buffer contains rhodamine only. The interneuron AIY is characterized by constant minute pulses which may be background measurement levels (typically <20% changes), or because it constantly integrates inputs from many other neurons. However, when this neuron is genuinely activated in response to changes in diacetyl concentrations, it correlates the activity of the AWA neurons (Fig. 4f), and often reaches 100% increase in neural activity (Fig. 4e). Note for example that the most significant activity observed in the AIY interneuron (two small peaks at time 2.2 and 2.5 minutes), correspond to the 2 pulses that are also observed in AWA at these times, as they were recorded from the same animal. Each panel shows neural traces from multiple worms.

Thus, we conclude that the intricate neural dynamics in these three neurons cannot be attributed to rhodamine, but is rather specific to diacetyl gradients. These new control

experiments now appear as Suppl. fig S14, and its legend provides the full explanation given above.

2. In most of the supplementary videos, there is significant movement. These motion artifacts (which should match the strength of the odor) are likely to affect the calcium responses.

We have now analyzed the data to verify that motion does not affect the observed neural activity. For this, we calculated the correlation between the neural activity and the position of the worm's head (measured by the displacement of the AWA neuron along the worm axis). We find no correlation between these two parameters which exclude the possibility that motion somehow affects neural activity ($\bar{\rho} = 0.05$, $p=0.15$, Wilcoxon Signed Rank Test, $n=18$ experiments in total). Moreover, we now provide analyses for the specific supplementary movies: M1(a), M4(b), M5(c), and M7(d). The control figure below is now added as Suppl. Fig S15. It is clear from the traces of the four movies that head position throughout the movie has no correlation with the salient pulsatile dynamics.

3. Fig 1B: Max concentration used in step-function is 1.2 mM. Fig 1D: It is a linear function, max at 0.6 mM. Fig 3B: A sigmoid with a max of ~190 μ M or ~375 μ M. Fig 4A: Sinusoid with max close to 1 mM. Why are all these maxes different? Or, do matching them obscure the results presented herein?

The reason for the different maximal concentrations is simply because we always loaded the stimulus syringe with the same diacetyl dilution (10^{-4}), which corresponds to 1.15 mM. For the step function experiments, this is indeed the final maximal concentration. For the different gradient shapes and slope magnitudes, different final concentrations are reached by the end of the experiment depending on the course of the gradient.

Notably, we observe the pulsatile activity phenomena across a wide range of gradients and final concentrations. For example, one can stop the linear gradient experiment in the middle, to reach a final concentration of 0.3 mM, which is equivalent to the higher sigmoid concentration, and still observe the pulsatile activity. Similarly, taking only the first $\frac{1}{4}$ of this experiment will match the 0.15 mM concentration reached in a different sigmoid experiment. Thus, the same results will hold regardless of the final concentration.

In addition, we tested two different linear slopes which differed by 10-fold for the same time period. This actually necessitated loading the stimulus syringe with a ten-fold lower concentration (the only case where we started off with a different concentration). In these experiments, we find the same pulsatile activity to hold for maximal concentrations of 0.6 mM and 0.06 mM (Fig 1e).

In the step function experiments (Fig. 1b,c), we measured the response to a 1.15 mM step and found a single pulse. Similarly, Larsch et al, "A Circuit for Gradient Climbing in *C. elegans* Chemotaxis", tested the response to steps of various final concentrations: 115 nM, 1.15 μ M and 11.5 μ M. Their result was always a single pulse. Specifically, for the 1.15 μ M (a 1000-fold lower conc. than ours), a single pulse is observed even when imaging for 6 minutes following the step.

As for the experiments where we demonstrated adaptation to the first derivative (Fig. 3), indeed, final concentrations may be a relevant issue. For this reason, we controlled for same final concentrations with different maximal slopes, and for similar maximal slopes with different final concentration (Fig. 3). In all these experiments, we always observed the same result, where the neuron adapts to the magnitude of the 1st derivative.

As for the sinusoidal gradient, we extended the range of concentrations as much as possible, hoping to capture a variety of different responses for both AWC and AWA. Although the maximal concentration in the syringe was 1.15 mM, the operation mode of the syringes precluded reaching this exact value (but it did reach a very close one, ~1 mM).

To conclude, in all our experiments, the final concentration was never an issue, and, indeed, all our results hold for a wide range of concentrations tested. We now provide a detailed description of the technical considerations to generate different gradients and the possible final

concentrations to reach (in Supplementary Material pp. 2-4). We also reference Larsch et al to demonstrate that a single pulse is observed for much lower step stimuli (pp. 4).

4. Fig 1G: AWA in individual worms have unique pulsatile activity. Is this behaviorally meaningful? Optogenetics would be an ideal experiment to test this idea, but at least the mechanisms could be discussed.

The connection between AWA variability and its behavioral meaning is indeed interesting. Our results, using freely-moving worms, demonstrated the correlation between AWA activity and the propensity to move forward or to turn. Following the above suggestion, we have now analyzed directly how animals change behavior as a function of AWA activity. For this, we used the optogenetic tool Chrimson to directly manipulate AWA using light in freely-moving worms. Indeed, we found that worms are more likely to move forward when AWA is active, and turn when AWA activity drops (now appears as a new supplementary figure S4). This experiment clearly shows the causality between AWA and behavior (in fact, Larsch et al showed a similar result). Together, these results strongly suggest that the large variability between animals in AWA activity can be directly translated to large behavioral variabilities (as indeed is often observed when performing behavioral assays).

We now discuss this interesting possibility in the discussion (pp. 10):

“We observed a large variability among individual worms in the AWA response to smooth gradients (Fig. 1g). Our findings (Fig. 2, and supplementary fig. S4), as well as those obtained by Larsch et al³⁹, indicate that AWA activity dictates behavioral outputs (e.g., runs and turns). Thus, the high variability in AWA activity may underlie, and presumably explain, at least some of the extensive behavioral variability often observed in chemotaxis assays (see for example, chemotaxis assay reported in Itskovits et al. 2017).”

5. Fig 3B: were there any attempts to increase the concentration at the sigmoid midpoint, or the length of time to achieve it? Furthermore, the authors modulate 1) time and 2) concentration. Time and concentration are both changed in the orange sigmoid, however, and it does not seem like the authors had a stimulus where only the time was changed. it might be interesting to do that experiment by using the same concentration midpoint of the blue (or red) curve.

We have now performed an additional experiment with a fourth gradient that is two-fold steeper and consequently reaches the (same) final concentration significantly faster (new figure 3). Specifically, the red curve sigmoid reaches from 10% to 90% of the final concentration within 8.5 minutes, while in the newly added condition (purple curve), the time interval between these concentrations is only 3.3 minutes (even though both conditions cross the midpoint at exactly the same time). Similarly to all other conditions, the newly added sigmoid condition (purple) shows exactly the same result where neural activity adapts to the first derivative (n=15, p=6e-

05). We have now modified figure 3 to include the new experimental condition that further supports our conclusions.

Along similar lines, the maximal steepness (at the midpoint) of the yellow curve sigmoid is similar to the maximal steepness of the red and blue curves, yet they reach different max levels (maximal slope is 46 μ M/min for all three, as opposed to 124 μ M/min for the purple). Again, in all these cases, we find that AWA activity adapts to the magnitude of the 1st derivative.

6. Fig 4J: It is unclear to me that faster chemotaxis is better chemotaxis, since it decreases roaming and, hence, the ability to look for better food sources. Perhaps using shallower gradients would be a better test.

We have now added to the simulations a navigation in shallower gradients. Shallower gradients practically translate to lower signal to noise ratios, so we repeated the simulations while incorporating increasing noise levels. We find that in the presence of an increasing noise, animals that use the 1st derivative strategy always outperform animals using the classical biased random walk strategy, as they reach the target for a wider fraction of parameters (P+ and P-). Thus, the strategy of adapting to the 1st derivative is both faster and more reliable to reaching the target. This is now demonstrated in the newly added supplementary figure S13.

One may also consider the chances to reach the target as the major factor (and possibly superior to time to reach the target). Our simulations show that adaptation to the 1st derivative

also increases the probabilities that animals will reach the target for a wider range of turning probabilities (P+ and P- in our simulations). For example, in figure 5c, there is a space of parameters where only animals, exploiting the 1st derivative adaptation principle, reach the target, while animals that rely solely on the sign of the first derivative fail to do so (Fig 5c, red area in the right of the black line).

In general, the speed (time to reach the target) is one possible parameter to look at. Since our experiments and simulations always included a single cue, the ability to attend and locate this stimulus provides a good measure for the overall success/fitness: the first to reach the food target will enjoy it. Thus, when we refer to faster, this actually reflects animal chances to reach the target in a given time frame. Obviously, given an infinite time, any animal obeying to the simplest random walk would reach the target as well. In that sense, we think that faster is better.

7. The differential pulsing between AWAR and AWAL neurons is very interesting. I would recommend discussing these data in the context of the known neuroanatomical connections with downstream neurons.

We agree this is an interesting observation. We have now added to the discussion section our findings in the context of the neuroanatomical connections (pp. 11):

“An additional type of neural variability lies in the activity of the left-right, anatomically symmetric, AWAR and AWAL neurons. Interestingly, in some of the worms, either AWAR or AWAL responded to the gradients, while in other worms both neurons responded. Furthermore, even in cases where both neurons responded, they often differed in their activation patterns (Fig. 4 e-f, and supplementary fig S10). This differential dynamics may be particularly interesting in light of the neuroanatomical connections that these neurons make²⁰. When analyzing the available connectome, both neurons directly connect (either by chemical synapses or via gap junctions) to the first layer of interneurons (e.g., AIY, AIA and AIZ). However, unlike AWAL, AWAR is also synapsing onto deeper layer interneurons, namely, RIR, RIFR, and RIGR. This differential wiring may hint to variable outputs depending on the origin of the pulsing neuron (right or left). However, in our analyses, we could not systematically characterize a difference between the activation patterns of AWAR and AWAL.”

8. Fig S10: The authors were able to measure calcium transients in moving animals. Are there significant differences between these responses to ones obtained from animals that are anesthetized?

We have now analyzed the features of the pulses as observed in awake and anesthetized animals. We do not find significant differences in their amplitude ($p=0.37$) nor in their frequency ($p=0.23$, Wilcoxon Rank Sum Test). The figure below, which we now add as supplementary fig. S16, shows the distributions.

Since we couldn't reliably fit decaying exponentials to the awake worms, we did not include this parameter in the comparison.

9. Supp Fig S2: Do pulsatile characteristics change when using the mutant strains defective in neurotransmitter release or neuropeptide secretion?

We have repeated the analyses as shown for the comment above. The figures below provide the results which we now add as supplementary fig. S3. We find that *unc-31* has a longer decay time (by 41%, $p = 0.008$) and shorter peak to peak period (by 27%, $p=0.03$) than WT, and *unc-13* has a lower amplitude than the WT (by 18%, $p=0.001$). These appear to be minor differences that could be also due to the integrity of the AWA neuron itself, or could point to weak feedbacks from nearby neurons. Thus, the pulsatile activity is indeed cell autonomous, however, it could be weakly modulated by connected neurons. We now raise these observations in the results, pp. 4:

“While primarily cell autonomous, other neurons may be modulating this pulsatile activity, as the parameters depicting this activity slightly varied in the neurotransmitter and the neuropeptide defective animals (Supplementary fig. S3).”

10. Simulation: $M = 30$ memory steps. What happens if fewer are used? How robust is the novel chemotactic strategy to changes in this parameter? (For instance, the authors changed M to 50, and find similar results, but no other values seem to be tested)

We have now repeated the simulations for various memory lengths (shown in a newly added supplementary fig. S12). We find that even for a short memory length of $M=2$, there is already a significant $\sim 20\%$ improvement in directionality. Increasing the memory length further improves the directionality towards the target, although very mildly (by few percent). This improvement becomes especially important when P_+ is slightly higher than P_- (dark red stripe in heat map, fig. S12b). This is the area where maximal improvement ratio is reached when comparing two memory lengths. We have now added the figure below and the corresponding legend that depict these analyses and results.

Minor

1. Supp Fig S3: presumably, the worm strains used in Ref. 8, which were used to make this figure, are the same reported in the rest of this article. They should also be in similar environments.

The worm strain used in Ref. 8 is the wild type strain (N2), showing that worms turn even when they are oriented directly towards the target. The results in the revised version appear in supplementary fig. S5. In this study, to measure Ca activity we generated a transgenic line expressing GCaMP in AWA, hence this strain is regarded as WT in all aspects but obviously it is transgenic (indeed it responds to diacetyl and attracted to it). The experiments were indeed performed in similar environments: in both, the animals were placed on a chemotaxis agar (made using the same recipe), and a drop of diacetyl was placed few centimeters away to generate the gradient. Growth and worms' maintenance conditions were exactly the same in both works (herein and the one mentioned in Itskovits et. al, 2017).

2. Fig 3B: it looks like the orange curve was shifted back by 9 minutes, not 10 minutes.
Thank you for pointing out this typo. We have now fixed the figure legend accordingly.

3. The authors imaged neurons in the microfluidic device at a rate of 1.4 frames/sec. This may be too slow to reasonably capture the activity of the studied neurons, though it does seem that the dynamics they saw at this imaging rate were slow enough to be mathematically described.

The pulsatile response that we characterized in this study is on time scales of many seconds (each pulse typically lasts 20s-60s). Thus, the frame rate we were using 1.4 Hz is fast enough to capture these relevant properties. To verify that this frame rate is indeed sufficient for capturing the pulsatile dynamics, we have now imaged few animals with a higher frame rate (4 Hz). In the figure below, we show the activity dynamics for the original data points (sampled at 4 Hz, blue) and the dynamics for the same data but down-sampled to 1.4 Hz (red). As is evident, the activity captured with the lower, under-sampled, frame rate is in high agreement with the one captured with the higher frame rate and included all the fine minute pulsatile peaks. Since the pulses are on timescales of many seconds, measuring at 1.4 Hz faithfully depicts the intricate dynamics so that no activity data is lost, overlooked, or under-sampled due to this lower frame rate. We have now added this figure to the revised manuscript as supplementary fig. S17.

Reviewer #2 (Remarks to the Author):

Report:

First of all, my apologies for the delay in submitting my review.

Overall, I find this manuscript potentially interesting and intriguing, but for the reasons detailed below, I have the feeling it is an incomplete and, to some extent, a rushed piece of work. Also, in my opinion many important details are omitted and some specific facts are generously generalized. I thus encourage the authors to do MAJOR REVISIONS on the manuscript so as to be a solid, clear, and comprehensive research piece.

Title and abstract

Regarding the title, it reads too pompous: rather than announcing the results of a specific piece of research, it reads like the title of a classic book on the topic. What are those "principles"? The authors could write down specifically what they mean. What is that "neural coding"? One can be concrete and precise without losing generality. And, "for efficient navigation": well, it does not hurt to say for what organism, as the manuscript does not present results for many species. In sum, I think it is fair to the readers to say precisely what was found without making it sound more (nor less) than what it is.

We now added to the title that the neural coding is found in *C. elegans*. We found the title to be cumbersome when we tried to precisely describe the neural principle, so we preferred to keep the following title: "Principles of neural coding for efficient navigation in *C. elegans*"

Regarding the abstract, it seems to start with the usual "mantra" of "navigation is central to survival", and that is fine, but perhaps it is too much to say that "little is known about how animals navigate based on chemical cues". Perhaps there is still a lot to learn, but certainly a lot is known about chemotaxis, at many levels (behaviorally, molecularly, circuit-wise, etc) and in many organisms (bacteria, worms, larvae, and rodents).

We have now revised the abstract accordingly.

The abstract insists on a "previously unknown mechanism" and a "novel principle". I think the authors could make a more in depth scholarly effort to mention previous results, both at the behavioral and coding levels, where very similar mechanisms have been found (ie. Schulze et al eLIFE on larval coding of temporal olfactory traces, , tons of work by the Bargmann lab, sophisticated analysis on worm navigation by Samuel lab, Lockery, etc, the lino paper on weathervaning, also in the larva by Gomez-Marin and Louis, and even in rodents by Bhalla amongst others).

In the Introduction section of the revised version, we now provide a detailed description of all these previous related works (pp. 2-3).

The abstract ends proposing that such mechanism is generalizable to other sensory modalities. Is it a rhetoric statement to try to increase generality of the findings, or is there a specific rational or evidence to claim that?

Given our findings that as few as three sensory neurons underlie this mechanism, we end the abstract with this intriguing possibility that other sensory modalities may employ similar mechanisms for navigation. In fact, following interactions with scientists studying various sensory systems, our findings inspired them to start searching for similar principles in their model systems.

Main text

As mentioned for the abstract, the authors perhaps underestimate in the first paragraph what is known about the neural and behavioral basis of chemotaxis.

We have now significantly modified the introduction. We now extensively elaborate on the previously known behavioral and neural correlates of chemotaxis in various organisms (pp. 2-3):

“Multicellular organisms, that harbor a neural system, use more sophisticated strategies when navigating based on chemical gradients. For example, chemotaxis of C. elegans nematodes is comprised of long periods of sinusoidal movement, termed ‘runs’, and intermittent turning events, where a bout of consecutive turns is known as a ‘pirouette’⁵⁻⁷. In their seminal work, Shimomura and Lockery⁷ demonstrated that C. elegans worms

modulate the probability to perform a pirouette based on the sign of the first derivative of the sensed stimulus⁷, similarly to the classical biased-random walk strategy observed in single-cell organisms. However, and unlike unicellular models, worms show a clear directional bias when exiting the pirouette: worms entering a pirouette following a run that was directed towards the target are more likely to exit the pirouette in the same goal-directed angle; Conversely, if entering a bout of turns following a run that was directed opposite to the target, then the exit angle is likely to be closer to 180 degrees, thus reorienting the animals towards the target^{7,8}. Later studies showed that, in addition to modulating pirouette rates based on the sign of the first derivative, worms also take into account the magnitude of the derivative⁸⁻¹⁰. In addition, animals use a second navigation strategy in chemical gradients, termed klinotaxis. In this strategy, animals continuously make smooth and gradual curvature corrections towards the chemical source, in a process termed in *C.elegans* ‘weathervane’¹¹⁻¹³.

To support such complex navigation maneuvers, neural circuits perform various computations. These include adaptation and temporal integration of the sensed concentrations^{6,14}; coding the magnitude of the change in the concentration¹⁴⁻¹⁶, bilateral coding^{17,18}, and temporal coding^{6,19}. To study coding principles and computations performed by individual neurons and circuits, it is useful to focus on animal models with a defined nervous system. In that respect, *C. elegans* nematodes offer a unique opportunity: It consists of a compact nervous system (302 neurons in total) for which a detailed wiring diagram is available²⁰. Indeed, studies characterized worm chemotaxis behavior^{7-9,21-23}, as well as the neural response to a variety of different stimuli²⁴⁻²⁶; Furthermore, recent advanced experimental systems measure neural activity in freely-behaving animals, allowing to infer the neural correlates of chemotaxis behavior^{16,24,27,28}.”

In the second paragraph, could the authors specify what information step-functions carry, beyond saying they "carry little".

As we have extensively modified the introduction in the revised version, we now also specify the information that step functions carry (pp. 3):

“Traditionally, chemosensory activity in *C. elegans* has been studied by presenting chemical cues in an on/off step-like manner, while simultaneously imaging activity from target neurons^{17,24,25,29-31}. These step-like stimulations may approximate turbulent plumes, where signals are patchily distributed, and animals are exposed to cues that rapidly fluctuate in time and space³²⁻³⁴.”

In the third paragraphs, they authors say "In agreement with previous reports," But there is no citation.

We now provide the citation for this on pp. 4:

“Similar single-pulse responses to a range of on-step levels of diacetyl were observed by others as well³⁹.”

The citation refers to Larsch et. al, 2015.

Also, I found the references regarding what is known about neural responses to temporally changing concentrations very scarce, poor and not making justice to the great progresses in the last 5 years or so, including previous pioneering work (in worms, but also in the larva, in locusts, and in rodents).

We have now extensively modified the manuscript and included a detailed introduction and discussion sections that describe and reference the results obtained in recent years (including the various model systems, e.g., worms, fly larvae, and rodents). We also describe studies which used chemical gradients as the experimental setup.

The development of the tracking system to enable measurement of neural activity in animals that freely-navigate through chemical gradients that they authors report is difficult to assess with respect to existing similar methods. Please provide more details and also discuss to what extent what is presented is novel or different. If it is not, that's fine too, but it must be specified.

We have now added a detailed description of our freely-moving animal neural imaging system in the methods section. Furthermore, we uploaded to github the full code for operating the system and for analyzing the movies.

<https://github.com/zaslab/FreelyMovingNeuronTracker/blob/master/Manual.pdf>

Briefly, we developed this tracking system in-house. It uses a regular epifluorescence microscopy system with a single camera and optical path. It is capable of both tracking a single worm moving on a plane, and reading its neural activity with a magnification of 10x. More advanced systems were already built. The major advantage of our system is that it is built on an off-the shelf microscope and does not require building ad-hoc microscopy setups. We now add this to the methods section as well (pp 14):

"Imaging freely behaving worms during chemotaxis

For these experiments, we developed a software package based on the Micromanager⁵² software suite for tracking and fluorescence imaging using a commercially available microscope setups. The code utilizes a motorized stage and a light source to track the worm while exciting and imaging its calcium sensor in 10X magnification. The code for this system, together with a detailed description of the entire system, can be found in our lab's github repository: <https://github.com/zaslab/FreelyMovingNeuronTracker>. In this study, we used an Olympus IX-83 inverted microscope, with a 10X UPLASAPO objective, Lumen-Dynamics' X-Cite light source, Prior H117 motorized stage and Photometrics Evolve 512 camera."

Can the authors justify and specify the figures of the "high-throughput chemotaxis assays" they mention.

In order to quantify the probability of a worm to perform a turn even when directed (as we observed in the freely moving worm, Fig. 2), we used data previously acquired (Ref 8 Itskovits et. al). Based on this data, we calculated the probability of a worm to turn given its bearing to the chemoattractant source. The previously acquired data here is a set of movies of experiments in which ~100 worms were placed on a petri dish together with a point source of a chemical attractant. The worms were then tracked with a previously described tracker (Ref 8, Itskovits et.al). From the acquired trajectories, and the position of the chemoattractant, we were able to extract turning events and their bearing to the chemoattractant source. Since each of these experiments yielded thousands of turning events, we regarded this as a high-throughput method. An example of an animal track and the angular statistics is found in Suppl. Fig S5.

We now provide all this information (and also refer to our recent publication describing these high-throughput assays) in the methods section (pp. 15):

“We used previously published data⁸ to analyze the turning rate of worms given their bearing in relation to the chemical source. In each of those experiments, approximately 100 worms were placed on one vertex of equilateral triangle with edge lengths of 4 cm, while diacetyl and dilution buffer were placed on the other two vertices. Tracking was done using the Multi Animal Tracker software suite⁸.“

We read that "Naturally-formed chemical gradients are typically Gaussian", but in many cases turbulence may disturb those gradients, or simply their own diffusion will make them change in time. None of these possibilities seem to be contemplated. Also, it would be nice to have a reference or some arguments to better justify that worms indeed often encounter quasi-steady Gaussian chemical gradients. Note that then the authors concentrate on sigmoidal functions.

We now provide a detailed explanation in the Introduction for why we used gradients and how this is relevant to *C. elegans* ecology (pp. 3):

“Traditionally, chemosensory activity in *C. elegans* has been studied by presenting chemical cues in an on/off step-like manner, while simultaneously imaging activity from target neurons^{17,24,25,29-31}. These step-like stimulations may approximate turbulent plumes, where signals are patchily distributed, and animals are exposed to cues that rapidly fluctuate in time and space³²⁻³⁴. However, animals, particularly small-size animals, are often found in limited and confined environments. For example, *C. elegans* worms are frequently recovered from rotting fruits³⁵, which constitute a secluded and turbulent-free environment, where abrupt changes in concentrations are presumably uncommon. In such settings, stable gradients may be formed due to diffusion from bacterial microenvironments or food deteriorating signals³⁶. These gradients are expected to be smooth and continuous due to simple spatiotemporal diffusion processes.”

We specifically used tanh sigmoidal gradients, rather than Gaussian, since we also wanted to focus on the adaptation to the 1st derivative, so the function had to be symmetric in terms of the 1st derivative. We now explain this in the results section (pp. 6):

“To test whether coding of the first derivative magnitude is time invariant, we designed our fine-controlled microfluidic system to generate a sigmoidal (namely, hyperbolic tangent) gradient (Fig. 3ai), where the first derivative of the concentration monotonically increases to reach its maximum value exactly at the midpoint of the gradient, after which the first derivative values symmetrically decrease (Fig. 3aii).”

The "novel principle" their "findings point to" is certainly not novel beyond the worm. For instance, Schulze et al eLIFE 2015 explored a wide range of temporal gradients and showed, empirically and by means of analytic modeling and numerical simulations, that a single olfactory sensory neuron can adapt to changed in concentration, differentiate and integrate them, which, given the wide ranges of sensory protocols used in that study, and the very properties of their model, clearly showed that fly larvae detect and use information beyond the first temporal derivative. None of this is mentioned by the authors, nor regarding previous worm data.

Indeed, the work done by Schulze et al discovered interesting coding principles showing that neurons can perform various computations (e.g., integrate and differentiate gradient signals). The important new finding in our study is that a neuron can also adapt to the first derivative of the gradient (as opposed to all previous studies that report on adaptation to the absolute concentrations of the stimulus). We now clearly specify that the novel principle refers to this phenomenon of adaptation to the first derivative in the results (pp. 6):

“Together, these findings point to a novel principle, whereby neural activity adapts to the magnitude of the first derivative of the gradient.”

And also in the discussion (pp. 9):

“In the present study, we revealed a novel type of adaptation: the neuron can also adapt to the magnitude of the gradient first derivative (Fig. 3). We show that this adaptation is beneficial as it allows worms to constantly search for trajectories with increasing first derivatives, therefore choosing trajectories better directed towards the target (Fig. 5)”

Could the authors clarify why the so called "pulsative activity" is not simply noise? I do not see from their plots or their movies, how one would be confident that such signals is faithfully used by the worm to navigate. I am sorry if that was obvious from the presented results, but it was not obvious to me.

Several observations clearly demonstrate that the observed pulses are genuine neural responses and not noise:

1. In the different supplementary movies (for example, movie M3), one can deduce the basal level of the neuron fluorescence and the fluctuations around it (normally, on the scale of 1-5% of the basal level signal). In contrast, the pulses that begin as soon as the worms encounter the gradient are more than 10-fold higher than the noise level, where the overall neural activity increases by between 50-200%. This shows that the neural pulses are elicited by the diacetyl gradients and are not an imaging artifact.
2. We now added controls comparing AWA neural responses to a sinusoidal gradient with and without diacetyl (with rhodamine in both cases, see supplementary fig. S14). This clearly shows that the AWA neurons are not activated in response to rhodamine, possible pressure changes or other sources of noise. However, in the presence of the diacetyl gradients, the mean fold change fluorescence significantly increases (Wilcoxon rank sum test, $p < 10^{-4}$).
3. This is further strengthened by figure 1b,c which clearly shows that in the presence of a constant diacetyl concentration, AWA does not exhibit a long pulsatile activity, but only a single pulse aligned to the stimulus onset. Thus, multiple pulses are observed only in gradients. A constant concentration of diacetyl (even after the step) does not elicit any response by itself.
4. Figure 2 shows that the observed pulses carry relevant information regarding the worm's behavior. In fact, these pulses dictate behavioral outputs and therefore cannot be regarded as mere noise: We found that while AWA fluorescence signal rises, the worm stays directed towards the target without turning. However, following its fall, worms tend to reorient. Indeed, we find that the mean change in direction is significantly lower in the 40 seconds before the pulse maximum compared to 40 seconds after it (Wilcoxon rank sum test, $p = 0.0054$). Additionally, comparing the maximal deviation in the same time windows shows that the maximal deviation is significantly higher after the pulse maximum (Wilcoxon rank sum test, $p = 0.0035$). Thus, pulses are the neural correlates of the behavioral outputs.
5. We now added experiments in which we use optogenetics to activate the AWA neurons. From these experiments, we find that when activating the AWA neurons during a run, the probability that the worm will stop the run and perform a turn is significantly lower than if not activating the neuron ($p < 10^{-10}$, supplementary fig. S4, supplementary movie M3). This shows a causation effect between AWA activity and the forward movement, again underscoring the importance of AWA activity in dictating worm's behavior.

We now stated all these points throughout the revised manuscript to make clear that the neural dynamics is a genuine response which carries information, and therefore cannot be regarded as noise.

In Figure 1, could the authors clarify what they mean at the end of the figure legend by "as if each animal maintains a typical individuality in pulsatile response". If that is the case, I find this pretty interesting.

Indeed, we also found the individuality to be compelling. In the figure legend, we now explain that each animal showed a characteristic pulsatile activity which significantly differed among the different animals (pp. 19):

“Pulses originating from the same animal were significantly more similar than pulses measured from other animals ($p < 10^{-6}$), as if each animal maintains a typical individuality in pulsatile response.”

We also added a paragraph in the discussion section that discusses the possible behavioral outcomes in light of these inter-individual variabilities (pp. 10):

“We observed a large variability among individual worms in the AWA response to smooth gradients (Fig. 1g). Our findings (Fig. 2, and supplementary fig S4), as well as those obtained by Larsch et al³⁸, indicate that AWA activity dictates behavioral outputs (e.g., runs and turns). Thus, the high variability in AWA activity may underlie, and presumably explain, at least some of the extensive behavioral variability often observed in chemotaxis assays (see for example, chemotaxis assay reported in Itskovits et al. 2017).”

Figure 2, where it is claimed that "the pulsative activity dictates behavioral outputs", one can see an example of a worm trajectory plus its neural activity, which is said to recapitulate the data observed in the microfluidic device, but I do not see how faithful and extrapolable that is to claim that the pulsative activity dictates navigation. In what sense?

Indeed, to extrapolate and, moreover, to show the direct causality, we have now added new experiments in which we optogenetically controlled AWA activity in freely-behaving worms. We found that worms are significantly more likely to continue moving forward when AWA is active, and make a turn as soon as the light switched off and AWA activity decreased ($p < 1e-10$). These results are now summarized in a new figure provided below (Supplementary Fig. S4), and we discuss this in the Results section (pp. 5):

“To further understand how AWA activity modulates worm behavior, we used Chrimson^{39,46} to light-activate the AWA neuron. We found that in times that AWA was active, turns were significantly suppressed ($p < 10^{-10}$, X^2 test, supplementary fig S4, and supplementary movie M3). This result is in agreement with Larch et al, who additionally found that immediately after AWA activation, turning probability increased above baseline³⁹. Taken together, these results demonstrate that AWA pulsatile activity dictates forward movement in times that AWA is rising, and a turn once its activity decays.”

In Figure 3, I wonder why the temporal scales are so large, namely, of the order of minutes. Plus, these are mainly representative/illustrative plots, but no statistical claims are presented.

Since the temporal scale of a single pulse is on the order of tens of seconds, the relevant time scale to observe multiple pulses and their adaptation will be on the order of minutes. Importantly, we observed the adaptation to the first derivative (shown in figure) 3 by comparing the two flanking 2.5-minutes time intervals around the midpoint of the gradient. In addition, these are non-spiking neurons which show relatively slow and gradual activation (unlike spiking neurons).

We now provide a better explanation for these timescales in the discussion (pp. 11):

“Similarly to classical adaptation processes that support sensitivity to a wide range of signal intensities⁴⁹, here, adaptation to the magnitude of the first derivative allows animals to code a broad range of derivatives when seeking shorter paths towards the source. Indeed, to support a robust coding in face of a wide range of derivatives, this adaptation occurs on relatively long timescales, typically, several AWA pulses which correspond to dozens of seconds (Fig. 3). This longer timescale extends the single bout of forward locomotion (a ‘run’) and encompasses a longer chemotaxis period in which the worm learns through adaptation the expected first derivatives in its surrounding environment.”

As of the statistical inference, we provided the results for the entire cohort of tested animals in one of the gradients (Fig 3a). We also analyzed the difference between the activities before and after the maximal first derivative value (2.5 flanking minutes, averaged over time) for all the gradients that we tested. The difference between those values is statistically significant (Signed Ranked Sum) for all the tested gradients (4 gradients included in the new version of the

manuscript). We have now made these significance values apparent in the figure body, figure legend, and in the text.

Similarly, in figure 4, we see some recapitulation of results in panel (e) for the correlation coefficient, but all the rest is essentially illustrative plots. Moreover, could the authors better specify what they mean by "orchestrated neural dynamics"?

This figure is indeed overloaded with experimental data, simulations, and an illustrative model that captures the premises of this study. In the current revised version, we split this figure into three (now figs 4,5,6), so that each is focused on one main issue and hence receives a better and clearer description. Thus, the new figure 4 in the revised version depicts AWA and AWC neural responses to a sinusoidal gradient with the following statistics:

- 1) AWA activity is lower during the decreasing phases of the gradient (Signed Wilcoxon Rank test, $p=0.0085$, $n=12$.)
- 2) AWC activity is higher during the decreasing phases of the gradient (Signed Wilcoxon Rank test, $p=0.00012$, $n=7$.)
- 3) There is a significant difference in correlations between AWA and AWC, showing that the AWC neurons respond deterministically compared to the stochastic pulsatile activity of AWA (Wilcoxon Ranked Sum test, $p<10^{-7}$, $n=21$).
- 4) We also show in figure 4 the low correlations between AWAR and AWAL relatively to the correlation between the AWA pair and the AIY pair.

The simulations part now appears in figure 5, showing that using the strategy with first derivative adaptation allows more directed chemotaxis than the classical biased random walk.

We end with figure 6 that summarizes the new findings in this work in one illustrative model. The term orchestrated neural dynamics is aimed to explain how the timely interplay between two sensory neurons, that exhibit completely different dynamics, provides an efficient navigation strategy.

We now better describe what we mean by orchestrated as we summarize this section in the Results, pp. 7:

“Taken together, when moving up a gradient, AWA and AIY pulsatile activity promotes forward movement, while reduced AWC^{ON} activity suppresses turning events^{17,31,42}. Conversely, when moving down the gradient, AWA activity is significantly lower ($p<0.009$, Wilcoxon Signed Rank test, Fig. 4d), thereby reducing prospects for forward movement, while AWC activity rises to promote turning events. “

In addition, the new figure 6 describes the navigation mechanism which is based on orchestrated dynamics of the two sensory neurons. The legend for figure 6 explains this intricate orchestrated dynamics (pp 28):

“(c) AWA and AWC dynamics in response to the sensed gradient. AWA stochastic pulsatile activity controls the worm forward movement. AWA adapts to the first derivative

of the gradient, so when the gradient begins to flatten (and although the first derivative is still positive), AWA activity decays, thereby promoting a turn. In case the turn steers the worm away from the target, the negative gradient is immediately sensed by the AWC neuron, which responds robustly and deterministically to promote a second turn. Thus, the orchestrated activity of AWA and AWC underlies the efficient navigation strategy in chemical gradients.”

We also provide a broader view on the interplay between pairs of neurons in *C. elegans*, in the discussion part, pp. 9:

“Interestingly, studies in *C. elegans* worms demonstrated how an interplay between two sensory neurons may have functional roles in coding environmental cues^{6,17}. For example, the anatomically homologues ASEL and ASER gustatory neurons sense NaCl, where ASEL is stimulated by increases in NaCl concentration, and ASER is stimulated by decreases in NaCl concentration. Behaviorally, ASEL activity prolongs forward locomotion whereas ASER promotes turning events¹⁷. In an analogous manner, ASH neurons respond to an increase in the repellent odorant 2-nonanone, leading to a bout of turns, while the AWB neurons respond to a decrease in odor concentration, leading to turn suppression⁶. Thus, orchestrated dynamics of two sensory neurons may be a common design to efficiently integrate environmental signals before relaying the information to drive the appropriate behavioral outputs.”

The simulations on chemotaxis performance are interesting but somewhat limited. Could the authors show the effect of noise in the integration, and also the kernels that do so, and the typical timescales necessary and plausible in the past?

Following this suggestion, we have now performed new simulations, this time adding noise to the system. In each simulation step, a random noise perturbation was drawn from a Gaussian distribution with mean 0 and a constant standard deviation. Several different values of standard deviation were tested and compared (See Supplementary fig. S13). As seen from the results, the higher the noise the worse is the performance. Adding low levels of noise (which are small relative to the values of the mathematical gradient of the Gaussian at the starting point), have little effects on chemotaxis performance, and high noise levels greatly impair performance of both chemotaxis strategies. In all cases tested, adding noise does not change the fact that navigating with the ability to adapt to the first derivative is superior to the classical biased random walk strategy.

To examine the dependence of our model on memory length, we repeated the simulations for various memory lengths (Supplementary fig. S12). For simplicity, a uniform kernel was used, giving the same weight to all experienced first derivatives. From the results, we found that increasing memory length indeed improves performance, and even short memory of $M=2$ provides a significant performance advantage (20%). Moreover, having a longer memory

enables successful chemotaxis for a larger range of P+ and P- values (Supplementary Fig. S12b).

We now provide all these figures in the new revised manuscript with the explanations of the various considerations.

Also, very little is discussed about the AWA neurons, and its relation to previous works and situation in known circuits.

We have now provided a detailed description about AWA and AWC neurons, and their known role in the worm's sensory system and chemotaxis (Introduction section, pp. 3-4):

“The odorant diacetyl is one example for a signal secreted from bacteria in rotting fruits³⁹. C. elegans worms strongly attract to this odorant, as it potentially indicates food sources. Diacetyl is sensed by two pairs of amphid sensory neurons in the worm - AWA and AWC²⁵. Furthermore, AWA is the only neuron that expresses the diacetyl GPCR, ODR-10, which is required for chemotaxis towards low concentrations (<10 μ M) of diacetyl²⁶. Behaviorally, AWA activity has been shown to suppress turning events³⁸, while AWC activity is correlated with reversals¹⁶. In response to an abrupt increase in diacetyl levels, AWA activity transiently increases and eventually adapts to the new concentration, in a history-dependent manner³⁸. In contrast, AWC responds to an off step in diacetyl²⁵. Both AWA and AWC are connected to first-layer interneurons (e.g., AIY and AIA) that control worm navigation^{38,40-42}.”

Supplementary Information

In the Supplementary Online Information I found just hints of crucial issues related to materials and methods.

First, in section 2, could the authors clarify if they really mean that the "developed microfluidic-based system allows generated ANY DESIRED smooth temporal function of a gradient", or, namely, to be quantitative about what is meant by "smooth".

We have now rephrased this statement to say that our system can generate a wide range of smooth gradients (pp. 4). We also provided a detailed discussion that elaborates on the parameters that dictate the possible smooth gradients in the supplementary (pp. 2-3):

"Let $r(t)$ be the flow rate function of the syringe pump which carries the chemical of interest. We use a total constant flow rate R , meaning that the buffer syringe pump flow

rate is $R-r(t)$. We denote V to be the mixing chamber volume and $F(t)$ to be the volume of odorant solution (in μL) inside the mixing chamber. The change of $F(t)$ in time is given by the difference between the odorant entering the mixing chamber and the odorant exiting it:

$$\frac{dF(t)}{dt} = r(t) - R \cdot \frac{F(t)}{V}$$

Here $r(t)$ is the odorant volume entering the mixing chamber every second, and $R \cdot \frac{F(t)}{V}$ is the amount leaving the chamber, as R is the total volume leaving the chamber and $\frac{F(t)}{V}$ is the fraction of the volume in the mixing chamber originating from the odorant syringe.

As we are interested to control the concentration of the odorant exiting the mixing chamber (which eventually enters the microfluidic device), we need to find the flow rate that will provide $C(t) = \frac{F(t)}{V}$. Hence, the pump rate should be:

$$r(t) = \frac{dF(t)}{dt} + R \cdot \frac{F(t)}{V} = V \cdot \frac{dC(t)}{dt} + R \cdot C(t)$$

This provides the flow rates needed for generating the desired function. For example, a linear temporal gradient is described by $C(t) = a \cdot t$ and therefore:

$$r(t) = V \cdot a + R \cdot a \cdot t$$

Our system parameters include V (the chamber size) and R , the constant flow rate to the chamber. These parameters dictate the maximal instantaneous change in concentration ($\frac{dC}{dt}$) that the system can support. We will derive the relationship between these parameters, and the maximal concentration change. The maximal flow $r(t)$ is bounded by R thus:

$$\frac{dC(t)}{dt} \cdot V + C(t) \cdot R \leq R \Rightarrow \frac{dC(t)}{dt} \leq \frac{R(1 - C(t))}{V}$$

On the other hand, $r(t)$ cannot be negative:

$$\frac{dC(t)}{dt} \cdot V + C(t) \cdot R \geq 0 \Rightarrow \frac{dC(t)}{dt} \geq \frac{-R \cdot C(t)}{V}$$

Thus, to achieve a larger instantaneous concentration change rate, one can increase the constant flow rate or decrease the chamber's volume.

Moreover, it is clear from the derivation above that the upper and lower bound on $\frac{dc}{dt}$ are concentration dependent. If $C(t) \approx 1$ (mixing chamber is almost full with odorant) then $\frac{dc}{dt}$ is bounded from above to be close to 0. And on the other hand, for $C(t) \approx 0$, $\frac{dc}{dt}$ is bounded from below by $\frac{-R \cdot C(t)}{V} \rightarrow 0$, which means the slope of the gradient has to become shallower as it approaches 0. Thus, as the gradient is approaching its maximal and minimal point, there are strict limitations on $\frac{dc}{dt}$. Due to these constraints, we avoided generating gradients that rise fast near the maximal concentration or drop fast near 0. For example, our sinusoidal gradient only reached a maximal point of 80% ($\max(C) = 0.8$) from the maximal concentration in the syringe. In our experiments (Figs. 1,3,4), we used flow rates ranging between $R = 25 \frac{\mu l}{sec}$ and $R = 150 \frac{\mu l}{sec}$ and mixing chambers with volumes ranging between $V = 50 \mu l$ and $V = 200 \mu l$.

Are any of the terms in the "Modeling system dynamics for generating temporal gradients" based on laws for fluid dynamics and, if so, could the authors specify which ones and why.

Our mathematical model does not take into consideration fluid dynamics. We now describe thoroughly the validity of our mathematical model in face of fluid dynamics, and their possible effect on the gradient (Supplementary material pp. 3-4):

"In addition to the theoretical constraints of the model, there are also practical constraints. In order for our mathematical description to hold, we assumed that the flow rate, set by the syringe pump, is the same flow rate to enter the mixer. For this to hold we had to avoid expansions due to flow pressure in the syringes or the tygon tubing that connects the syringes to the mixer. To allow this, we used a low resistance microfluidic chip (a wide and short tunnel), which requires low pressures to operate, and minimized tubing length from the syringes to the mixing chamber. Most importantly, we used glass syringes to reduce possible expansion due to the building pressure inside. Another requirement is that the fluid in the mixing chamber will have enough time to mix within it before it leaves towards the microfluidic chip. Assuming few turns of the magnetic stirring bar are enough to uniformly mix the fluids, this requires stirring at a rate of ~2 hz to allow smoothing of our minutes-long gradients. This rate was easily achieved using a standard commercial magnetic stirrer. Another point for consideration is the diffusion

and turbulent processes during and along the flow inside the tube which may cause a small amount of the odorant to arrive prior to the expected timing based on calculations. This results in a neural response which may be observed ~1 minute ahead of its expected timing. An example of such a case is shown in Figure 1d.”

I am no expert in "imaging neural activity", and so I cannot comment in detail, but I presume some other experienced reviewer may want to know more about the details of it.

In the revised version, we did our best to provide all the possible details to better describe our imaging system. We also added the following parameters to the methods describing the imaging parameters: pinhole opening (1.2 Airy units), z slice jumps (0.7 μm) and recording temperature (20 Celsius).

Regarding section (4), Imaging freely behavior worms during chemotaxis, I have several questions. To prevent external perturbations of the gradient, the authors use a coverslip above the imaged worm. From experience, I know that simply closing the cover slip can cause turbulence and therefore disturb the odor gradients. Could the authors comment on that and also provide quantitative evidence that the gradients are stable.

In the revised manuscript, we provide quantitative evidence to show that the formed gradient is stable. We elaborate and discuss the effect of the coverslip on the formed gradient in the Methods section, pp. 16:

“Given the diffusion constant of diacetyl is $D \approx 9 \frac{\text{mm}^2}{\text{sec}}$ (calculated based on its molecular mass), it should take roughly $t = \frac{L^2}{D} = 30 \text{ sec}$ for the gradient to stabilize once the coverslip is placed. The worm was kept in the $5 \mu\text{L}$ droplet for approximately 3 minutes before the droplet evaporated, giving the diacetyl gradient enough time to stabilize.”

The freely-moving worm is imaged at 2Hz, which I think is not quite enough. Given the typical timescales of worm locomotion (body bending) and head movements, it has been standard for many years now to monitor at least at 15Hz, if not faster. And this is no big deal nowadays. Having 2 frames per second may miss much of the postural changes of the worm, with which it implements locomotion (including the "wide-angle head swings during movement" that the authors mention), and thus navigation, as well as miss the fast timescales related to the motor-sensory loops and perceptual dynamics from the perspective of the animal.

Indeed, a 2 Hz imaging rate may not capture all the fine details of the worm behavior during chemotaxis (e.g., the rapid head swings, or the specific characteristics of the undulations). However, in this study, we focused on the AWA pulsatile activity which occurs on a timescale of

at least several seconds, and so is the behavioral correlates of AWA activity. Specifically, we presented here the effect of AWA activity on the transition between “runs” and intermittent “turns” which are behavioral modes on the scales of several seconds. These behaviors are clearly observed and accurately analyzed into details using our 2 Hz tracking system. Even more so, this frame rate also sufficed to reliably capture neural dynamics in the freely moving worms.

In fact, in image analyses we considered the animal as a point on the plane and disregarded the fine details (such as head swings) in our subsequent analyses. For this, we actually used 2D smoothing algorithm to eliminate the high frequency undulations of the worm movement, but accurately preserve the trajectory. We now add Supplementary fig. S18 to demonstrate the raw data prior to smoothing (below), and also explicitly specify how we extracted the data (Methods, pp. 14-15):

“We then analyzed the movies using custom-made MATLAB software to extract neuron activity together with the worm position in relation to the chemical source. For accurate determination of worm trajectories, and to compensate for the wide-angle head swings during movement, we smoothed worm tracks (Supplementary fig. S18) with a smoothing spline, that uses a least-squares approach with penalization for roughness^{54,55}.”

The custom-made worm tracker lacks all details. Please explain something about it. Similarly, the Matlab scripts used to control the tracker, track the worm and extract neuron activity should be made available in established repositories (ie. sourceforge or github) and/or included as supplementary material. Otherwise, it is a black box.

As we described for a previous comment, we have now added a detailed description of our freely-moving animal neural imaging system to the methods section (pp. 14). In addition, we uploaded the code to github:

<https://github.com/zaslab/FreelyMovingNeuronTracker/blob/master/Manual.pdf>

For the multi-worm tracker, we cite our recent paper (Itskovits et al 2017) where we detail on its features and also provide its entire code and documentation in github (url is in the revised version).

Regarding data analysis on the behavioral side, the only specification we read in the supplementary material is "we smoothed worm tracks". This is not enough. What filters? What time windows, etc, etc?

We now describe the smoothing algorithm we are using in the methods section (page 16):

“For accurate determination of worm trajectories, and to compensate for the wide-angle head swings during movement, we smoothed worm tracks (Supplementary fig. S18) with a smoothing spline, that uses a least-squares approach with penalization for roughness 53,54”

And of-course, referencing the actual algorithm that was used:

53 Garcia, D. Robust smoothing of gridded data in one and higher dimensions with missing values. *Comput Stat Data An* 54, 1167-1178, doi:10.1016/j.csda.2009.09.020 (2010).

54 Garcia, D. A fast all-in-one method for automated post-processing of PIV data. *Exp Fluids* 50, 1247-1259, doi:10.1007/s00348-010-0985-y (2011).

I found Point (5), the Simulating chemotaxis performance, interesting and valuable. Yet, again, the authors seem to think that details are irrelevant. For instance, given the Gaussian gradient they simulate, what were its mean and variance? And, did it evolve in time, as physical diffusion may imply?

We have now gathered all these parameters and they appear in the same paragraph, under the Stimulating chemotaxis performance section (Supplementary material, pp. 8):

“The gradient function was: $C(r) = e^{\frac{-r^2}{2\sigma}}$, with $\sigma = 100$ [au²]

Speed: 1/simulation step [au]

Starting distance from source: 300 [au]

Stop distance: this parameter determines at what distance from the point source the simulation will stop: 30 [au].

Stop time: this parameter determines after how many time steps the simulation will stop if the stop distance was not yet reached: 3000 steps.

Memory length (M): 30 steps”

In addition, we did not evolve the Gaussian gradient in time according to the diffusion equation but kept it constant. Indeed, we now emphasize this in the supplementary information (pp. 8):

“the gradient was kept constant during the entire simulation and did not evolve in time according to the diffusion equation.”

In the simulations, time goes as $t=T+1$, but can the authors specify what is the dt , and whether timescales matter with respect to real physical and biological units of the worm?

Our simulations intended to theoretically examine the possible benefits that arise from adapting to the magnitude of the experienced first derivative. Therefore, we simplified our model as much as possible, without trying to directly simulate ‘real life’ time scales necessary for its implementation.

We now explicitly explain this in the manuscript (methods section, pp. 11):

“However, herein, we deliberately kept our simulations as general as possible as their sole purpose was to contrast between the two navigation strategies. Instead, we varied the various parameters of the model to show that regardless of their specific values, the strategy that employs the first derivative adaptation is always superior to the classical biased random walk strategy.”

However, we now discuss the appropriate scales that could be associated with our simulations in order to fit to the typical environment of the worm (Supplementary material, “Chemotaxis Simulations”, pp. 11):

“In an effort to provide an intuitive understanding of the simulation parameters and their relatedness to the typical scales of *C. elegans*, one may consider the following: A typical chemotaxis assay places the worm ~ 5 cm away from the source³. In our simulations we used 300 steps as the distance of the virtual animal from the source, implying that each

step is $\frac{1}{6}$ mm \sim 200 μ m. Average worm speed is 0.2 mm/sec⁴, thus each time step (dt) corresponds to \sim 1 second.”

Note that using the sign of the temporal changes in concentration assumes some sort of normalization of concentration baseline. The authors could comment on previous evidence for that.

Indeed, observing the neural response to the derivative suggests adaptation to the absolute odorant concentration, as observed in previous works as well. We now explicitly elaborate on these studies in the discussion (page 9):

“AWA neurons show two different modes of adaptation to external gradients of diacetyl. In the first mode, the neurons adapt to absolute levels of diacetyl, allowing them to remain sensitive to any changes in the concentration (Fig 1. b-c, and ³⁸). This type of adaptation is used by many organisms as part of the biased random walk chemotaxis strategy^{38,47,48}. In the present study, we revealed a novel type of adaptation: the neuron can also adapt to the magnitude of the gradient first derivative (Fig. 3). We show that this adaptation is beneficial as it allows worms to constantly search for trajectories with increasing first derivatives, therefore choosing trajectories better directed towards the target (Fig. 5).”

If I recall correctly, EColi does reorient randomly in any direction (and in 3D). As for worms, this may not be so (please double-check classical work by Shimomura and Lockery). This is relevant since in the simulation the authors "chose the new direction randomly from a uniform distribution".

Indeed, worms do not uniformly sample a new direction following a turn. Shimomura and Lockery et al. had shown, and we confirmed this in a previous work (Itskovits et.al 2017). However, the goal of the simulations was to compare between two strategies: a simple run&tumble biased random walk, and one that also applies adaptation to the first derivative magnitude. We therefore aimed to keep our simulation as simple and general as possible. In addition, by not accounting for the specific features relevant for *C. elegans*, the results and the conclusions could be generalized to other animals as well.

We now explain in the discussion our simulation approach and the possible conclusions that can be drawn from it, in light of previous simulation approaches (pp. 10):

“Previous navigation studies also used simulations to provide a model that recapitulates the observed behavior. These type of simulations incorporated specific behavioral

characteristics in order to reproduce the fine navigation features that were experimentally observed^{6,12,14}. However, herein, we deliberately kept our simulations as general as possible as their sole purpose was to contrast between the two navigation strategies. Instead, we varied the various parameters of the model to show that regardless of their specific values, the strategy that employs the first derivative adaptation is always superior to the classical biased random walk strategy.”

It is very important to discuss further the role and values of the "memory length" M (it is only mentioned later in the text that $M=30$ is used, and that they "observe the same general behavior" for $M=50$, but what does that mean?), cause it may point to the integration properties of the worm (see for instance the discussion on the explicit quantitative model for Transient Normalization, including adaptation, normalization and differentiation in the *Drosophila* larva by Schulze et al. eLIFE).

We have now performed thorough simulations, where we systematically changed the memory parameter, M . based on these new simulations we found that increasing M improves chemotaxis directionality and especially helps in regimes where P_+ is close to P_- (see also our response to Reviewer 1). We also found that even a short memory of $M=2$ is sufficient to greatly improve directionality compared to the classical biased random walk. We now added these analyses as a new supplementary figure S12.

“Supplementary figure S12 | Increasing memory length slightly improves chemotaxis performance. We repeated the described simulations in Figure 5 for several values of the memory length M . (a) The mean projection for increasing lengths of memory as calculated over all values of P_+ and P_- for which at least a single animal was able to reach the target using the classical biased random walk strategy. Red line marks the mean

projection calculated for the classical biased random walk strategy. It is evident that increasing the memory length somewhat improves directionality, but even for $M=2$, a substantial increase over the classical biased random walk strategy is observed. (b) Fold-improvement in the performance between $M=32$ and $M=64$. Black line marks the $P_+ = P_-$ line. The fold-improvement is defined as $100 \cdot (M=64 \text{ mean projection} - M=32 \text{ mean projection}) / (M=32 \text{ mean projection})$ For most values of P_+ and P_- fold-improvement is very small, but at the chemotaxis limit, around the black line, fold improvement increases as worms with a shorter memory length can no longer reach the target.”

And in the results (pp. 8):

“Importantly, the superiority in performance is insensitive to the parameters chosen, as similar results are obtained for a wide range of parameters, for example, when varying the adaptation memory time, or when adding noise to the gradient sensed by the animal, (Supplementary figs. S12-S13).”

Also, note that all physical and biological variables (ie. worm speed, distance to source, gaussian decay, time, etc) are [au], which means arbitrary units, but to have some plausibility, the authors should try to map them into the well-known typical scales of the worm.

As we explained for the above comment, the simulations were not intended to model *C. elegans* typical scales, but rather meant to be general with the sole goal to provide comparison of two strategies. For this reason, we now covered a wide range of parameters, such as turning probabilities given a positive or negative gradients, memory length and noise. In addition, we used a scale-free linear gradient which allows to disregard specific diffusion parameters and also to analytically solve chemotaxis performance.

However, in an effort to provide some plausible relevant units for *C. elegans*, we derive the following parameters: A typical chemotaxis assay places the worm ~ 5 cm away from the source. In our simulations we used 300 steps as the distance of the ‘animal’ from the source, implying that each step is $\frac{1}{6}$ mm ~ 200 μ m. Average worm speed is 0.2 mm/sec, thus each dt corresponds to ~ 1 second.

We now provide these considerations in the “Chemotaxis Simulations” procedure in the supplementary (pp. 9):

“In an effort to provide an intuitive understanding of the simulation parameters and their relatedness to the typical scales of *C. elegans*, one may consider the following: A typical

chemotaxis assay places the worm ~ 5 cm away from the source. In our simulations we used 300 steps as the distance of the virtual animal from the source, implying that each step is $\frac{1}{6}$ mm ~ 200 μ m. Average worm speed is 0.2 mm/sec, thus each time step (dt) corresponds to ~1 second.

The authors, when discussing their results on the projection score say that "we surprisingly find... meaning it is beneficial to occasionally reorient even when moving up the gradient". Well, that is not surprising at all, isn't it? If the animal moving up the gradient, it can still increase its navigation efficiency by aligning better to the local gradient. This has been shown clearly, for instance, in *Drosophila* larvae (Gomez-Marin et al, Nature Communications 2011), and it makes sense for any organisms that would want to improve its chemotaxis.

We have now rephrased our statement to better explain what we meant.

Indeed, *Drosophila* larvae tend to redirect to more directed trajectories when climbing up the gradient. This strategy makes perfect sense when the reorientation event is indeed biased towards a better bearing in reference to the chemical source.

However, our simulations show that even when choosing from a uniform distribution, where most chances are to turn into a worse direction, then this strategy is still better than turning only when facing negative gradients.

Our simulations show that this strategy will become efficient only if this it is coupled with a rapid correcting mechanism, such that the animal will make a turn soon after it encounters a negative gradient. Indeed, this is the mechanism that we also find for *C. elegans*, in which AWC robustly promotes turns in case of negative gradients.

We now better explain this in the supplementary ("Chemotaxis Simulations", pp. 7):

"This may be surprising since according to our simulations, in each turn, the worm's new direction is chosen randomly from a uniform distribution, meaning that most chances are to turn into a less oriented trajectory. However, we find that for high values of P_- , when the cost associated with a wrong turn is sufficiently small, taking these 'risks' may actually become beneficial (Supplementary fig. S11a)."

Also I find missing in the discussion the more subtle chemotaxis phenomenology known for the worm, such as the lino paper on weathervaning (and its counterpart in the larva by Gomez-Marin and Louis in 2014).

We now added this important information both in the introduction and the discussion sections:

In the introduction (pp. 2):

“In addition, animals use a second navigation strategy in chemical gradients, termed klinotaxis. In this strategy, animals continuously make smooth and gradual curvature corrections towards the chemical source, in a process termed in C.elegans ‘weathervane’¹¹⁻¹³.”

And in the discussion (pp. 10):

“This efficient navigation strategy we report herein, where animals adapt to the first derivative of the gradient, joins other complex navigation strategies employed by multicellular organisms. These include a turn bias, where animals are likely to exit a pirouette better oriented towards the target^{7,8}, and klinotaxis, where animals make gradual curvature corrections towards the target¹¹⁻¹³. “

Finally, from the simulations, the authors find out that including information of the first derivative improves the classical biased-random walk strategy. And this is what they call the "principle". If so, the authors should carefully review and report the literature, in worms and other organisms, to report where such beyond-random-walk strategies have been discovered or discussed empirically, numerically and theoretically.

The principle we are pointing out in this study describes a novel, previously uncharacterized, process of adaptation to the first derivative of the gradient. We now better highlight the novelty and discuss it in light of all the previous relevant studies in the field:

In the introduction, page 2, we now describe other chemotaxis strategies:

“Multicellular organisms, that harbor a neural system, use more sophisticated strategies when navigating based on chemical gradients. For example, chemotaxis of C. elegans nematodes is comprised of long periods of sinusoidal movement, termed ‘runs’, and intermittent turning events, where a bout of consecutive turns is known as a ‘pirouette’⁵⁻⁷. In their seminal work, Shimomura and Lockery⁷ demonstrated that C. elegans worms modulate the probability to perform a pirouette based on the sign of the first derivative of the sensed stimulus⁷, similarly to the classical biased-random walk strategy observed in single-cell organisms. However, and unlike unicellular models, worms show a clear directional bias when exiting the pirouette: worms entering a pirouette following a run that was directed towards the target are more likely to exit the pirouette in the same goal-

directed angle; Conversely, if entering a bout of turns following a run that was directed opposite to the target, then the exit angle is likely to be closer to 180 degrees, thus reorienting the animals towards the target^{7,8}. Later studies showed that, in addition to modulating pirouette rates based on the sign of the first derivative, worms also take into account the magnitude of the derivative⁸⁻¹⁰. In addition, animals use a second navigation strategy in chemical gradients, termed klinotaxis. In this strategy, animals continuously make smooth and gradual curvature corrections towards the chemical source, in a process termed in *C.elegans* ‘weathervane’¹¹⁻¹³.”

And in the Discussion (pp. 10):

“This efficient navigation strategy we report herein, where animals adapt to the first derivative of the gradient, joins other complex navigation strategies employed by multicellular organisms. These include a turn bias, where animals are likely to exit a pirouette better oriented towards the target^{7,8}, and klinotaxis, where animals make gradual curvature corrections towards the target¹¹⁻¹³. “

Finally, and insisting on a minimum of ecological relevance, the authors say that a 2D exponential decay gradient (which is a Gaussian one, actually, rather than an exponential) "may better resemble genuine gradients animals are likely to encounter in nature". Can the authors provide some references to what is known in worms regarding such gradients?

We have now fixed this mistake in the revised version and state that we used a Gaussian (and not exponential) gradient. Moreover, we have also included in the simulations a linear gradient that showed the same results. We now added a paragraph describing when we expect to find Gaussian-shaped gradients, in the Introduction (pp. 3):

“However, animals, particularly small-size animals, are often found in limited and confined environments. For example, *C. elegans* worms are frequently recovered from rotting fruits³⁵, which constitute a secluded and turbulent-free environment, where abrupt changes in concentrations are uncommon. In such settings, stable gradients may be formed due to diffusion from bacterial microenvironments or food deteriorating signals³⁶. These gradients are expected to be smooth and continuous due to simple spatiotemporal diffusion processes.”

And in the results (pp. 5):

“The gradient formed by a single odorant source, in a non-fluctuating environment, is typically Gaussian, and animals navigating towards this source are likely to encounter a gradient with increasing first derivatives”

Finally, in the section (7) Data analysis, could the authors give more details as to the relevance of bootstrapping from 10 worms data to 1 million random shuffles?

In order to bootstrap the various pulses and show individuality, we shuffled between all pulses gathered from all 10 worms. Each worm was assigned with a random set of pulses but with the same number of pulses it originally had. These are the number of pulses originally assigned for each worm:

Worm index	1	2	3	4	5	6	7	8	9	10	SUM
Pulses number	9	17	9	9	4	7	5	9	11	12	92

From this, we could calculate the total number of possible permutations as:

$$\binom{92}{9} \cdot \binom{92-9}{17} \cdot \dots \cdot \binom{92-80}{12} \approx 10^{81}$$

Thus, the total number of random possible assignment of pulses is much larger than 1 million.

We now describe the bootstrapping process more thoroughly in the methods section (pp. 16, first paragraph):

“To analyze the variability of the pulsatile responses, a total of 92 discrete pulses were compiled from 10 different worms responding to a linear gradient (Fig. 1g) which depict responses from 6 of the worms). To test whether each worm is characterized by a significantly different pulse properties, we first calculated the standard deviations of different pulse parameters (namely amplitude, and pulse decay time) for each worm. We then shuffled all 92 pulses between the 10 worms, thus assigning each worm a random set of pulses, but each worm consisted with the same number of pulses it originally had. For this random set, we calculated the mean standard deviations for each of the pulse parameters and compared it to the mean standard deviations obtained for the original data. The results of this bootstrap analysis showed that the standard deviations of the

random shuffles (N=10⁶ in total) are significantly higher than those of the original data (p≤10e-6). "

Therefore, I encourage the authors to take these suggestions into consideration in order to strengthen the content, clarity and significance of their work.

In other to increase transparency in the delicate and important process of reviewing our peers' work, I always disclose my name: Alex Gomez-Marin.

END

Reviewer #3 (Remarks to the Author):

Itskovits et al., have used *C. elegans* as a model system to understand the mechanism of efficient navigation in chemical gradients. To do this they have analyzed chin response to external chemical stimulation. They have revealed that two chemosensory neurons, AWA and AWCON, have distinct responses, with AWA showing pulsatile responses in shallow odorant gradients and AWCON showing robust responses. Based on those experimental results, they use computational modeling to show how these two distinct functional chemosensory responses empower a more efficient navigation strategy than the classical biased random walk strategy. The manuscript is well-written and easy to read, and I think it is of high significance and appropriate for Nature Communications. A few experimental concerns are listed below.

The authors show (Fig 1d) clearly different responses to a step change of diacetyl vs a gradient from 0 to 0.6mM. It would be interesting to know what leads to this discontinuity. Have the authors tried much shallower gradients to see what the threshold for pulsatile responses is?

The discontinuity from a single to multiple pulses is indeed interesting; In that respect, it is interesting to find the minimal duration of varying stimulus concentration that will elicit a train of pulses. Larsch et. al 2013, imaged AWA activity in response to quick sharp gradients that reached a constant value within 5 seconds. This resulted in a single pulse, similar to what we observed for an instantaneous step function, suggesting that the gradients need to be longer than 5 seconds. Our data show that AWA responds with several pulses when facing continuous and much longer gradients. Each pulse has a time scale of tens of seconds. Thus, to elicit a pulsatile response, the duration of the varying stimulus should be on the time scale of several pulses, typically, more than one minute. Accordingly, when we presented the worm with a linear gradient, the pulsatile nature of the response becomes apparent after ~2 minutes (Fig. 1d). We

speculate that adaptation processes within the AWA neurons govern the transition from a single to multiple pulses. These adaptations last more than 5 seconds but less than 2 minutes.

Regarding the minimal slope from which pulsatile activity is observed, we have now analyzed our data to infer the threshold for the pulsatile response. As the tanh gradients started off with very low slopes, we focused at these early time points to detect when the first pulse appears. For example, in the figure below the first pulse appears at ~150 s (marked with a red circle). According to our concentration estimation (based on rhodamine, red channel) this corresponds to 0.002uM/sec. This is the lowest threshold that we detected among the very early responding animals (N=8 in total). We also provide the threshold slopes for the other 7 worms, where we find that the average is ~ 0.01 uM/sec (although with a considerable variability among the individuals). Interestingly, Larsch et al used step functions with extremely low concentrations, and found that AWA responded with a significant activity to a step as low as 0.0115 uM, and very moderately to 0.00115 uM. These similar values suggest that the AWA respond to changes as low as ~ 0.01 uM/sec.

Of note, since we extract these numbers based on extremely shallow slopes, which are close to the noise level of this specific experimental measurement, our inferred thresholds should be considered as the upper limit of the threshold only. Worms might actually be more sensitive than that.

Also, in Figure 1d there appears to be a response in AWA before there is a change in diacetyl concentration. Is this a misprint, or is there an explanation for this apparent anticipatory response?

We now explain the source of this early response in the revised version of the manuscript.

The observed 'early' response is due to diffusion, and possibly other fluid-flow, processes that affect only the start of the experiment. It takes several minutes from the start of the experiment, when we turn on the pumps and the diacetyl begins to flow out of the syringes, until the front of the diacetyl flow reaches the worm's nose in the microfluidic device. During this time, the diluted diacetyl diffuses ahead of the diacetyl flow front. Other flow processes, for example, minute tubing expansion (and which we have not integrated into our flow model) may also have an effect. Such processes result in low levels of diacetyl that reach the worm earlier than expected. For this reason, in some of the experiments, we observe a pulse before we could actually detect rhodamine levels.

Notably, this is only a start-of-the-experiment effect, which does not affect the gradients to follow throughout the experiment. As soon as a detectable amount of rhodamine molecules enter the system, we can reliably read its levels and infer the concentration at any given second (Fig. 1a).

We now explain this technical issues in the methods (pp. 12):

“Of note, diffusion, and possibly other fluid-flow processes in the tubing, causes a minute amount of the cue to arrive before its expected time based on calculation. This results in a neural response which may be observed up to 1 minute ahead of its expected time. An example of such a case can be seen in Figure 1d. Importantly, this is only a start-of-the-experiment effect, which does not affect the gradients to follow during the experiment. As soon as detectable levels of rhodamine enter the field of view, we can reliably quantitate them and accurately infer diacetyl concentrations at any given second (Fig. 1a).”

And in the Supplementary (pp. 4):

“Another point for consideration is the diffusion, and possibly other processes during and along the flow inside the tubing, which may cause a small amount of the odorant to arrive prior to the expected timing based on calculations. This results in a neural response which may be observed ~1 minute ahead of its expected timing.”

For Figure 1e, it was not clear from the text whether authors believe there is a difference between the responses observed in the $10e^{-8}$ to $10e^{-4}$ gradient vs the $10e^{-8}$ to $10e^{-5}$ gradient. If they believe there is a difference, they might want to plot out the data in a way that shows this more convincingly. In any case some clarification would be helpful.

Indeed, we now clarify this issue. We did not try to suggest such a difference (and it would not be right to do so) due to the large variability between animals within each of the conditions precludes such comparison (**Fig 1g**). For this reason, we compared neural activity in response to different slopes within the same animal by presenting a single worm gradually changing 1st derivatives. From these better controlled experiments (varying slopes for the same animal), we find that the higher the 1st derivative the higher is the amplitude and the shorter is the peak-to-peak time interval (shown in supp. figure S7).

We explicitly explain this in the results, bottom of page 5:

“As pulsatile activity is highly variable between animals (Fig. 1g), we analyzed neural responses of individual animals while exponentially increasing the gradient first derivative over time.”

In Figure 2a, is the software used for freely-moving imaging available for other researchers? It seems like a useful tool so this might be something to consider.

We have now added to the Methods section a detailed description of our freely-moving animal neural imaging system. Furthermore, we have uploaded to github the software code as well as a detailed manual for operating the system and for analyzing the movies. Available for download from: <https://github.com/zaslab/FreelyMovingNeuronTracker/blob/master/Manual.pdf>

Methods section pp. 14:

“The code for this system, together with a detailed description of the entire system, can be found in our lab’s github repository: <https://github.com/zaslab/FreelyMovingNeuronTracker>.”

The authors use a t-test to compare imaging results (e.g. third sentence, para 9). It might be worth also running a non-parametric test as the results may not be Gaussian in distribution.

We have now reanalyzed these results using non-parametric tests. The same results remain significant, and thus, do not change any of the conclusions. We now note this in the Results section, pp. 7:

“However, AIY activity decayed significantly faster than that of AWA ($t_{1/2} = 1.4$ sec vs 2.1 sec, respectively, $p < 10^{-8}$ Wilcoxon rank-sum test)”

Moreover, we have now used non-parametric tests, where appropriate, throughout the manuscript.

In Figure 3 the authors make the interesting claim that the neurons sense the first derivative of the chemical gradient rather than adapting to the prolonged exposure to the stimulus. However, it is possible that the first derivative only indicates the increment of concentration increase. To

see if it is truly the first derivative that is being sensed, the authors could conduct similar experiments with a sigmoidal decreasing concentration of odorant.

Indeed, in Figure 3 we show that during an increasing gradient, AWA not only senses the magnitude of the first derivative, but also adapts to it. We now emphasize this in the discussion (pp. 9, first paragraph).

“AWA also adapts to the first derivative, thus promoting a turn when the gradient begins to flatten, in search of a trajectory that is better oriented towards the target. “

Observing this effect in a decreasing gradient could indeed be very interesting. In fact, our dataset contains decreasing gradients (the decreasing phases in the sinusoidal gradient, **Fig. 4 a-c**). However, we observed very weak AWA activity during this decreasing period of the gradient (significantly weaker than in the increasing periods, **Fig. 4d**), from which we could not deduce meaningful conclusions. Similarly, previous reports did not observe AWA activity in response to a decrease in diacetyl concentration (Zaslaver et.al 2015, Larsch et.al 2015).

Thus, we do not expect to derive a meaningful conclusion regarding the first derivative during decreasing gradients.

Reviewers' comments:

Reviewer #1 (Remarks to the Author):

The authors have done a good job in answering my concerns. I have a minor point about the new AWA imaging data. I would have thought that the authors would optogenetically activate AWA in a pulsatile manner rather than a simple 10-second activation. This would have strengthened their data and provided more evidence to the conclusion.

Reviewer #2 (Remarks to the Author):

The authors have carefully addressed all my (many!) comments and suggestions: better literature, much more detail (tracking, stimulus delivery protocols, high-throughput, fluid dynamics, resolution), adding optogenetics, commented further on the issue of individuality, extended numerical simulations, improvement of figures, adding extra supplementary information. I believe the manuscript is now much more complete and precise.

I have fished two little "errors": (i) note that plots in the figures have labels that seem to have different fonts and that are stretched, (ii) when citing the navigation strategies in references 12-14 ('weathervaning', etc), the lino et al paper (ref 13) is indeed for the worm, but the other two are for *Drosophila* larvae, but the text says "worms".

One more petition: in the data availability, I believe that saying "all relevant data are available from the authors" defeats the purpose: to make it available is to make it available, rather than saying it would be available upon request. Thus I encourage the authors to publish it with the paper (either in NatComm or in dedicated repositories such as Dryad).

Two major concerns remain on my side:

First, I still think the title is too general and probably purposefully (but unjustifiedly) grandiloquent. "Principles of neural coding for efficient navigation in *C. elegans*" is too much.. The authors have add "*C. elegans*" to their previous title, and said they could not find a simple way to describe the neural principle they report, and thus left the title as is. Yet, again, I find this misleading in two respect. First, they highlight one computational principle in the manuscript, not principles (in plural). Second, there should be a way to say what principle that is. If it consists in the fact that a neuron adapts to the first derivative of the gradient, why not say that. Perhaps with the help of the editorial team, the authors could find a way to say what they find in their manuscript, no less and no more.

Second, I insist that observing adaptation to the first derivative of the gradient is not new in other species. In Schulze et al., for the *Drosophila* larva, transient normalization indeed implies (experimentally and also via the quantitative mathematical modelling and predictions therein) that the first derivative of the stimulus acts in the denominator and so the neural response adapts to it. That is actually one of the components of the "Dynamical feature extraction at the sensory periphery" that the title of that paper reported. In the abstract we wrote that "[w]e find that OSNs can act as differentiators that transiently normalize stimulus intensity". Yet, if one looks at the paper carefully (and I admit that is hard because there is so much in it), right after equation (1) —in page 10— we explain its meaning: "The denominator of this hyperbolic relationship contains a scaling term $S(x, t)$ that normalizes the spiking activity by the short-term history of changes in the stimulus intensity dx/dt ". Namely, its temporal derivative. Later, in page 39, we further expand it: solving for "y" in

equation (11) yields in the denominator a normalization based on the recent history of the first temporal derivative of the stimulus. As we explained in the paragraph after the equation: "As expected, the QSSA solution is in excellent agreement with the results of the integration of the full ODE system (Figure 14). Given the values of the parameters (Table 1), the denominator of relationship (11) is mostly driven by the stimulus intensity for slowly evolving stimuli. The contribution of the convolution over the first derivative is significant for rapid and large changes of the stimulus intensity." On the whole, the whole point of that work (as shown in Figure 14, and also in Figure 4; plus then the connection to behavior in figure 5, etc) was to demonstrate how that single neuron could extract all those features of the stimulus time course and use them to guide behavior. But I do not want to be too picky here. Whether the authors want to insist that this is the first time ever that first-derivative adaptation has been found or not, is up to them and the editor. I think that their results for the worm are significant enough to deserve publication even if that claim is not made so pontently.

I thus recommend the paper for publication.

Reviewer #3 (Remarks to the Author):

I think the authors have done a very nice job responding to reviewer comments and have improved the manuscript with new experiments. I am supportive of publication in this form.

A point by point response letter

Black – the original letter with the comments

Blue – Our response to the comments

Highlighted text – Changes as appear in the modified manuscript.

Reviewers' comments:

Reviewer #1 (Remarks to the Author):

The authors have done a good job in answering my concerns. I have a minor point about the new AWA imaging data. I would have thought that the authors would optogenetically activate AWA in a pulsatile manner rather than a simple 10-second activation. This would have strengthened their data and provided more evidence to the conclusion.

We thank the reviewer for the positive feedback. Regarding the optogenetic activation of the AWA neuron: we have light-activated the neuron (via Chrimson) for durations that are in the same time-scale of the natural pulses observed in response to the diacetyl gradients. As we demonstrate in supplementary movie M3, multiple intermittent light activations lead to the same behavioral outputs: the worm continued forward movement as long as AWA was active, and the worm backed as soon the we turned off the light stimulus.

In addition, Larch et al ("A Circuit for Gradient Climbing in *C. elegans* Chemotaxis"), used a 20-seconds optogenetic activation and similarly observed a decrease in turning probabilities during light activation periods (just like we observed for the 10-second pulses). The results of these optogenetic experiments are in full agreement with the results obtained with the freely moving animals and further support the notion that AWA activity suppresses turning events and the drop in AWA activity promotes turns. Based on all these observations, we concluded that there is a causal relationship between AWA pulsatile activity and the worm behavior.

Reviewer #2 (Remarks to the Author):

The authors have carefully addressed all my (many!) comments and suggestions: better literature, much more detail (tracking, stimulus delivery protocols, high-throughput, fluid dynamics, resolution), adding optogenetics, commented further on the issue of individuality, extended numerical simulations, improvement of figures, adding extra supplementary information. I believe the manuscript is now much more complete and precise.

I have fished two little "errors": (i) note that plots in the figures have labels that seem to have different fonts and that are stretched, (ii) when citing the navigation strategies in references 12-14 ('weathervaning', etc), the lino et al paper (ref 13) is indeed for the worm, but the other two are for *Drosophila* larvae, but the text says "worms".

We thank the reviewer for this comment.

(i) We made sure the fonts match the entire manuscript style.

(ii) We have now changed the references to correctly describe the text:

“In addition, animals use a second navigation strategy in chemical gradients, termed klinotaxis¹²⁻¹⁴. In this strategy, animals continuously make smooth and gradual curvature corrections towards the chemical source, in a process termed in *C.elegans* ‘weathervane’¹³.”

One more petition: in the data availability, I believe that saying "all relevant data are available from the authors" defeats the purpose: to make it available is to make it available, rather than saying it would be available upon request. Thus I encourage the authors to publish it with the paper (either in NatComm or in dedicated repositories such as Dryad).

We have now uploaded our main dataset into an open scientific repository (<http://osf.io>). It can be found under the name of the paper. Moreover, we have now uploaded our main analysis code into GitHub. Both the code and the data repository are now referenced in the manuscript, page 17:

“**Data availability:** The mean fluorescence values of the AWA neuron and the measured gradients throughout the experiments are available at <https://osf.io/> (Under the name of the paper).”

Code availability: The code for imaging freely-moving animals as well as the code for the simulations can be found in the github repository: <https://github.com/zaslab/>. Any additional data information is available upon request.”

Two major concerns remain on my side:

First, I still think the title is too general and probably purposefully (but unjustifiedly) grandiloquent. "Principles of neural coding for efficient navigation in *C. elegans*" is too much.. The authors have add "*C. elegans*" to their previous title, and said they could not find a simple way to describe the neural principle they report, and thus left the title as is. Yet, again, I find this misleading in two respect. First, they hilghlight one computational principle in the manuscript, not principles (in plural). Second, there should be a way to say what principle that is. If it consists in the fact that a neuron adapts to the first derivative of the gradient, why not say that. Perhaps with the help of the editorial team, the authors could find a way to say what they find in their manuscript, no less and no more.

We have now changed the title to:

“A concerted deterministic and stochastic neural coding enables efficient chemotaxis in *C. elegans*”

This title provides an accurate and specific description of the paper premise.

Second, I insist that observing adaptation to the first derivative of the gradient is not new in other species. In Schulze et al., for the *Drosophila* larva, transient normalization indeed implies (experimentally and also via the quantitative mathematical modelling and predictions therein) that the first derivative of the stimulus acts in the denominator and so the neural response adapts to it. That is actually one of the components of the "Dynamical feature extraction at the sensory periphery" that the title of that paper reported. In the abstract we wrote that "[w]e find that OSNs can act as differentiators that transiently normalize stimulus intensity". Yet, if one looks at the paper carefully (and I admit that is hard because there is so much in it), right after equation (1) —in page 10— we explain its meaning: "The denominator of this hyperbolic relationship contains a scaling term $S(x, t)$ that normalizes the spiking activity by the short-term history of changes in the stimulus intensity dx/dt ". Namely, its temporal derivative. Later, in page 39, we further expand it: solving for "y" in equation (11) yields in the denominator a normalization based on the recent history of the first temporal derivative of the stimulus. As we explained in the paragraph after the equation: "As expected, the QSSA solution is in excellent agreement with the results of the integration of the full ODE system (Figure 14). Given the values of the parameters (Table 1), the denominator of relationship (11) is mostly driven by the stimulus intensity for slowly evolving stimuli. The contribution of the convolution over the first derivative is significant for rapid and large changes of the stimulus intensity." On the whole, the whole point of that work (as shown in Figure 14, and also in Figure 4; plus then the connection to behavior in figure 5, etc) was to demonstrate how that single neuron could extract all those features of the stimulus time course and use them to guide behavior. But I do not want to be too picky here. Whether the authors want to insist that this is the first time ever that first-derivative adaptation has been found or not, is up to them and the editor. I think that their results for the worm are significant enough to deserve publication even if that claim is not made so potently.

I thus recommend the paper for publication.

Following this remark, we have now modified several sentences in the manuscript highlighting that the novelty in our study is a mechanism comprised of a concerted dynamics of two neurons: one robust and deterministic while the other is stochastic and pulsatile (see the highlighted changes in the new Word file). We also emphasize that it was previously suggested that animals keep memory of previous derivatives of the gradient (on page 9):

"Interestingly, *drosophila* larvae were also found to integrate past derivatives of the gradient to modulate future neural responses¹⁵."

In addition, in the abstract:

We deleted the term 'novel' from the principle of the adaptation to the 1st derivative but added the term few rows afterwards to emphasize that the orchestrated mechanism is novel (see highlighted text).

And on page 6,

We changed:

“Together, these findings point to a novel intriguing principle, whereby...”

to

“Together, these findings point to an intriguing principle, whereby...”

Similarly, in the discussion, page 8:

“Remarkably, we found that the AWA chemosensory neurons code smooth gradients via pulsatile dynamics, and elucidated an intriguing principle, where ...”

On page 11, we deleted the words ‘novel coding’ and instead refer to the new concerted mechanism – the orchestrated activity of AWA together with AWC – that promotes efficient navigation.

“In summary, here we report of a novel coding mechanism that...”

Is changed to

“In summary, here we report of an intriguing mechanism that...”

Reviewer #3 (Remarks to the Author):

I think the authors have done a very nice job responding to reviewer comments and have improved the manuscript with new experiments. I am supportive of publication in this form.

Thank you.